# OPTQ: Accurate Post-Training Quantization for Generative Pre-trained Transformers

**Elias Frantar**[*]
IST Austria

**Saleh Ashkboos**
ETH Zurich

**Torsten Hoefler**
ETH Zurich

**Dan Alistarh**
IST Austria & NeuralMagic

## Abstract

Generative Pre-trained Transformer models, known as GPT or OPT, set themselves apart through breakthrough performance across complex language modelling tasks, but also by their extremely high computational and storage costs. Specifically, due to their massive size, even inference for large, highly-accurate GPT models may require multiple performant GPUs, which limits the usability of such models. While there is emerging work on relieving this pressure via model compression, the applicability and performance of existing compression techniques is limited by the scale and complexity of GPT models. In this paper, we address this challenge, and propose OPTQ, a new one-shot weight quantization method based on approximate second-order information, that is both highly-accurate and highly-efficient. Specifically, OPTQ can quantize GPT models with 175 billion parameters in approximately four GPU hours, reducing the bitwidth down to 3 or 4 bits per weight, with negligible accuracy degradation relative to the uncompressed baseline. Our method more than doubles the compression gains relative to previously-proposed one-shot quantization methods, preserving accuracy, allowing us for the first time to execute an 175 billion-parameter model inside a single GPU for generative inference. Moreover, we also show that our method can still provide reasonable accuracy in the *extreme quantization* regime, in which weights are quantized to 2-bit or even *ternary* quantization levels. We show experimentally that these improvements can be leveraged for end-to-end inference speedups over FP16, of around 3.25x when using high-end GPUs (NVIDIA A100) and 4.5x when using more cost-effective ones (NVIDIA A6000). The implementation is available at `https://github.com/IST-DASLab/gptq`.

## 1 Introduction

Pre-trained generative models from the Transformer (Vaswani et al., 2017) family, commonly known as GPT or OPT (Radford et al., 2019; Brown et al., 2020; Zhang et al., 2022), have shown breakthrough performance for complex language modelling tasks, leading to massive academic and practical interest. One major obstacle to their usability is computational and storage cost, which ranks among the highest for known models. For instance, the best-performing model variants, e.g. GPT3-175B, have in the order of 175 billion parameters and require tens-to-hundreds of GPU years to train (Zhang et al., 2022). Even the simpler task of inferencing over a pre-trained model, which is our focus in this paper, is highly challenging: for instance, the parameters of GPT3-175B occupy 326GB (counting in multiples of 1024) of memory when stored in a compact float16 format. This exceeds the capacity of even the highest-end single GPUs, and thus inference must be performed using more complex and expensive setups, such as multi-GPU deployments.

Although a standard approach to eliminating these overheads is *model compression*, e.g. (Hoefler et al., 2021; Gholami et al., 2021), surprisingly little is known about compressing such models for inference. One reason is that more complex methods for low-bitwidth quantization or model pruning usually require *model retraining*, which is extremely expensive for billion-parameter models. Alternatively, *post-training* methods (Nagel et al., 2020; Wang et al., 2020; Hubara et al., 2020; Nahshan et al., 2021), which compress the model in one shot, without retraining, would be very appealing. Unfortunately, the more accurate variants of such methods (Li et al., 2021; Hubara et al., 2021; Frantar et al., 2022) are complex and challenging to scale to billions of parameters (Yao et al.,

---

[*]Corresponding author: `elias.frantar@ist.ac.at`

2022). To date, only basic variants of round-to-nearest quantization (Yao et al., 2022; Dettmers et al., 2022) have been applied at the scale of GPT-175B; while this works well for low compression targets, e.g., 8-bit weights, they fail to preserve accuracy at higher rates. It therefore remains open whether one-shot *post-training quantization* to higher compression rates is generally-feasible.

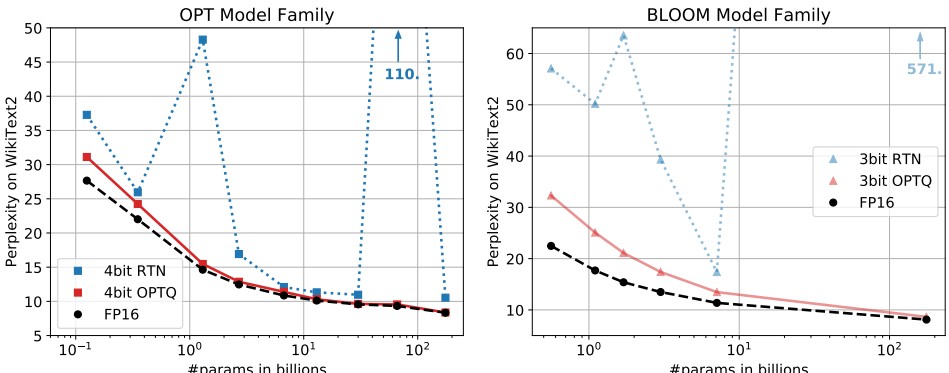

Figure 1: Quantizing OPT models to 4 and BLOOM models to 3 bit precision, comparing OPTQ with the FP16 baseline and round-to-nearest (RTN) (Yao et al., 2022; Dettmers et al., 2022).

**Contribution.** In this paper, we present a new post-training quantization method, called OPTQ,[1] which is efficient enough to execute on models with hundreds of billions of parameters in at most a few hours, and precise enough to compress such models to 3 or 4 bits per parameter without significant loss of accuracy. For illustration, OPTQ can quantize the largest publicly-available models, OPT-175B and BLOOM-176B, in approximately four GPU hours, with minimal increase in perplexity, known to be a very stringent accuracy metric.

Further, we show that our model can also provide robust results in the *extreme quantization* regime, in which models are quantized to 2 bits per component, or even *ternary values*. On the practical side, we develop an execution harness which allows us to execute the resulting compressed models efficiently for generative tasks. Specifically, we are able to run the compressed OPT-175B model for the first time on a single NVIDIA A100 GPU, or using only two more cost-effective NVIDIA A6000 GPUs. We also implement bespoke GPU kernels which are able to leverage compression for faster memory loading, resulting in speedups of $\approx 3.25\times$ when using A100 GPUs, and $4.5\times$ when using A6000 GPUs.

To our knowledge, we are the first to show that extremely accurate language models with hundreds of billions of parameters can be quantized to 3-4 bits/component: prior *post-training methods* only remain accurate at 8 bits (Yao et al., 2022; Dettmers et al., 2022), while prior *training-based* techniques have only tackled models that are smaller by one to two orders of magnitude (Wu et al., 2022). This high degree of compression may appear natural, as these networks are overparametrized; yet, as we discuss in our detailed analysis of results, compression induces non-trivial tradeoffs between the accuracy of the language modeling (perplexity), bit-width, and the size of the original model.

We hope that our work will stimulate further research in this area, and can be a further step towards making these models available to a wider audience. In terms of limitations, our method currently does not provide speedups for the actual multiplications, due to the lack of hardware support for mixed-precision operands (e.g. FP16 x INT4) on mainstream architectures. Moreover, our current results do not include activation quantization, as they are not a significant bottleneck in our target scenarios; however, this can be supported using orthogonal techniques (Yao et al., 2022).

## 2 RELATED WORK

Quantization methods fall broadly into two categories: quantization during training, and post-training methods. The former quantize models during typically extensive retraining and/or fine-tuning, using some approximate differentiation mechanism for the rounding operation (Gholami et al., 2021; Nagel et al., 2021). By contrast, post-training ("one-shot") methods quantize a pre-

---

[1]This merges the name of the OPT model family with the abbreviation for post-training quantization (PTQ).

trained model using modest resources, typically a few thousand data samples and a few hours of computation. Post-training approaches are particularly interesting for massive models, for which full model training or even finetuning can be expensive. We focus on this scenario here.

**Post-training Quantization.** Most post-training methods have focused on vision models. Usually, accurate methods operate by quantizing either individual layers, or small blocks of consecutive layers. (See Section 3 for more details.) The AdaRound method (Nagel et al., 2020) computes a data-dependent rounding by annealing a penalty term, which encourages weights to move towards grid points corresponding to quantization levels. BitSplit (Wang et al., 2020) constructs quantized values bit-by-bit using a squared error objective on the residual error, while AdaQuant (Hubara et al., 2021) performs direct optimization based on straight-through estimates. BRECQ (Li et al., 2021) introduces Fisher information into the objective, and optimizes layers within a single residual block jointly. Finally, Optimal Brain Quantization (OBQ) (Frantar et al., 2022) generalizes the classic Optimal Brain Surgeon (OBS) second-order weight pruning framework (Hassibi et al., 1993; Singh & Alistarh, 2020; Frantar et al., 2021) to apply to quantization. OBQ quantizes weights one-by-one, in order of quantization error, always adjusting the remaining weights. While these approaches can produce good results for models up to $\approx 100$ million parameters in a few GPU hours, scaling them to networks orders of magnitude larger is challenging.

**Large-model Quantization.** With the recent open-source releases of language models like BLOOM (Laurençon et al., 2022) or OPT-175B (Zhang et al., 2022), researchers have started to develop affordable methods for compressing such giant networks for inference. While all existing works—ZeroQuant (Yao et al., 2022), LLM.int8() (Dettmers et al., 2022), and nuQmm (Park et al., 2022)— carefully select quantization granularity, e.g., vector-wise, they ultimately just round weights to the nearest (RTN) quantization level, in order to maintain acceptable runtimes for very large models. ZeroQuant further proposes layer-wise knowledge distillation, similar to AdaQuant, but the largest model it can apply this approach to has only 1.3 billion parameters. At this scale, ZeroQuant already takes $\approx 3$ hours of compute; OPTQ quantizes models $100\times$ larger in $\approx 4$ hours. LLM.int8() observes that *activation outliers* in a few feature dimensions break the quantization of larger models, and proposes to fix this problem by keeping those dimensions in higher precision. Lastly, nuQmm develops efficient GPU kernels for a specific binary-coding based quantization scheme.

Relative to this line of work, we show that a significantly more complex and accurate quantizer can be implemented efficiently at large model scale. Specifically, OPTQ more than doubles the amount of compression relative to these prior techniques, at similar accuracy.

## 3 BACKGROUND

**Layer-Wise Quantization.** At a high level, our method follows the structure of state-of-the-art post-training quantization methods (Nagel et al., 2020; Wang et al., 2020; Hubara et al., 2021; Frantar et al., 2022), by performing quantization layer-by-layer, solving a corresponding reconstruction problem for each layer. Concretely, let $\mathbf{W}_\ell$ be the weights corresponding to a linear layer $\ell$ and let $\mathbf{X}_\ell$ denote the layer input corresponding to a small set of $m$ data points running through the network. Then, the objective is to find a matrix of quantized weights $\widehat{\mathbf{W}}$ which minimizes the squared error, relative to the full precision layer output. Formally, this can be restated as

$$\text{argmin}_{\widehat{\mathbf{W}}} \, ||\mathbf{W}\mathbf{X} - \widehat{\mathbf{W}}\mathbf{X}||_2^2. \tag{1}$$

Further, similar to (Nagel et al., 2020; Li et al., 2021; Frantar et al., 2022), we assume that the quantization grid for $\widehat{\mathbf{W}}$ is fixed before the process, and that individual weights can move freely as in (Hubara et al., 2021; Frantar et al., 2022).

**Optimal Brain Quantization.** Our approach builds on the recently-proposed Optimal Brain Quanization (OBQ) method (Frantar et al., 2022) for solving the layer-wise quantization problem defined above, to which we perform a series of major modifications, which allow it to scale to large language models, providing more than *three orders of magnitude* computational speedup. To aid understanding, we first briefly summarize the original OBQ method.

The OBQ method starts from the observation that Equation (1) can be written as the sum of the squared errors, over each row of $\mathbf{W}$. Then, OBQ handles each row $\mathbf{w}$ independently, quantizing one weight at a time while always updating all not-yet-quantized weights, in order to compensate for the error incurred by quantizing a single weight. Since the corresponding objective is a quadratic,

whose Hessian is $\mathbf{H}_F = 2\mathbf{X}_F\mathbf{X}_F^\top$, where $F$ denotes the set of remaining full-precision weights, the greedy-optimal weight to quantize next, which we denote by $w_q$, and the corresponding optimal update of all weights in $F$, denoted by $\boldsymbol{\delta}_F$, are given by the following formulas, where quant$(w)$ rounds $w$ to the nearest value on the quantization grid:

$$w_q = \operatorname{argmin}_{w_q} \frac{(\operatorname{quant}(w_q) - w_q)^2}{[\mathbf{H}_F^{-1}]_{qq}}, \quad \boldsymbol{\delta}_F = -\frac{w_q - \operatorname{quant}(w_q)}{[\mathbf{H}_F^{-1}]_{qq}} \cdot (\mathbf{H}_F^{-1})_{:,q}. \tag{2}$$

OBQ quantizes weights iteratively using these two equations, until all the weights of $\mathbf{w}$ are quantized. This is done efficiently, avoiding expensive full recomputations of $\mathbf{H}^{-1}$, by removing the $q$th row and column of $\mathbf{H}$, which is necessary after quantizing $w_q$, directly in the inverse via one step of Gaussian elimination. Namely, the updated inverse is given by the formula

$$\mathbf{H}_{-q}^{-1} = \left(\mathbf{H}^{-1} - \frac{1}{[\mathbf{H}^{-1}]_{qq}}\mathbf{H}_{:,q}^{-1}\mathbf{H}_{q,:}^{-1}\right)_{-p}. \tag{3}$$

This method comes with a vectorized implementation, handling multiple rows of $\mathbf{W}$ in parallel. Eventually, the algorithm can achieve reasonable runtimes on medium-sized models: for instance, it can fully quantize the ResNet-50 model (25M parameters) in $\approx 1$ hour on a single GPU, which is roughly in line with other post-training methods achieving state-of-the-art accuracy (Frantar et al., 2022). However, the fact that OBQ's runtime for a $d_{\text{row}} \times d_{\text{col}}$ matrix $\mathbf{W}$ has *cubic* input dependency $O(d_{\text{row}} \cdot d_{\text{col}}^3)$ means that applying it to models with billions of parameters is extremely expensive.

## 4 THE OPTQ ALGORITHM

**Step 1: Arbitrary Order Insight.** As explained in the previous section, OBQ quantizes weights in greedy order, i.e. it always picks the weight which currently incurs the least additional quantization error. Interestingly, we find that, while this quite natural strategy does indeed seem to perform very well, its improvement over quantizing the weights in arbitrary order is generally small, in particular on large, heavily-parametrized layers. Most likely, this is because the slightly lower number of quantized weights with large individual error is balanced out by those weights being quantized towards the end of the process, when only few other unquantized weights that can be adjusted for compensation remain. As we will now discuss, this insight that *any fixed order may perform well*, especially on large models, has interesting ramifications.

The original OBQ method quantizes rows of $\mathbf{W}$ independently, in a specific order defined by the corresponding errors. By contrast, we will aim to quantize the weights of *all rows in the same order*, and will show that this typically yields results with a final squared error that is similar to the original solutions. As a consequence, the set of unquantized weights $F$ and similarly $\mathbf{H}_F^{-1}$ is always the same for all rows (see Figure 2 for an illustration). In more detail, the latter is due to the fact that $\mathbf{H}_F$ depends only on the layer inputs $\mathbf{X}_F$, which are the same for all rows, and not on any weights. Therefore, we have to perform the update of $\mathbf{H}_F^{-1}$ given by Equation (3) only $d_{\text{col}}$ times, once per column, rather than $d_{\text{row}} \cdot d_{\text{col}}$ times, once per weight. This reduces the overall runtime from $O(d_{\text{row}} \cdot d_{\text{col}}^3)$ to $O(\max\{d_{\text{row}} \cdot d_{\text{col}}^2, d_{\text{col}}^3\})$, i.e., by a factor of $\min\{d_{\text{row}}, d_{\text{col}}\}$. For larger models, this difference consists of several orders of magnitude. However, before this algorithm can actually be applied to very large models in practice, two additional major problems need to be addressed.

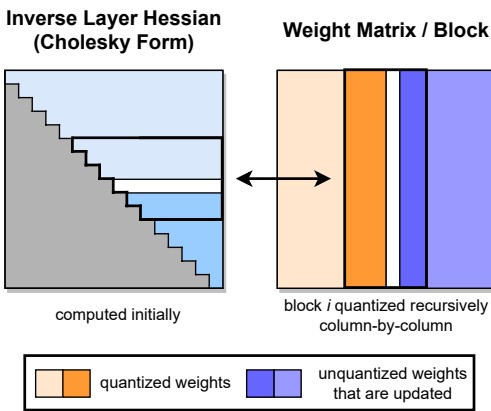

Figure 2: OPTQ quantization procedure. Blocks of consecutive *columns* (bolded) are quantized at a given step, using the inverse Hessian information stored in the Cholesky decomposition, and the remaining weights (blue) are updated at the end of the step. The quantization procedure is applied recursively inside each block: the white middle column is currently being quantized.

**Step 2: Lazy Batch-Updates.** First, a direct implementation of the scheme described previously will not be fast in practice, because the algorithm has a relatively low compute-to-memory-access ratio. For example, Equation (3) needs to update all elements of a potentially huge matrix using just a

few FLOPs for each entry. Such operations cannot properly utilize the massive compute capabilities of modern GPUs, and will be bottlenecked by the significantly lower memory bandwidth.

Fortunately, this problem can be resolved by the following observation: The final rounding decisions for column $i$ are only affected by updates performed on this very column, and so updates to later columns are irrelevant at this point in the process. This makes it possible to "lazily batch" updates together, thus achieving much better GPU utilization. Concretely, we apply the algorithm to $B = 128$ columns at a time, keeping updates contained to those columns and the corresponding $B \times B$ block of $\mathbf{H}^{-1}$ (see also Figure 2). Only once a block has been fully processed, we perform global updates of the entire $\mathbf{H}^{-1}$ and $\mathbf{W}$ matrices using the multi-weight versions of Equations (2) and (3) given below, with $Q$ denoting a set of indices, and $\mathbf{H}^{-1}_{-Q}$ denoting the inverse matrix with the corresponding rows and columns removed:

$$\boldsymbol{\delta}_F = -(\mathbf{w}_Q - \text{quant}(\mathbf{w}_Q))([\mathbf{H}_F^{-1}]_{QQ})^{-1}(\mathbf{H}_F^{-1})_{:,Q}, \tag{4}$$

$$\mathbf{H}^{-1}_{-Q} = \left(\mathbf{H}^{-1} - \mathbf{H}^{-1}_{:,Q}([\mathbf{H}^{-1}]_{QQ})^{-1}\mathbf{H}^{-1}_{Q,:}\right)_{-Q}. \tag{5}$$

Although this strategy does not reduce the theoretical amount of compute, it effectively addresses the memory-throughput bottleneck. This provides an order of magnitude speedup for very large models in practice, making it a critical component of our algorithm.

**Step 3: Cholesky Reformulation.** The final technical issue we have to address is given by numerical inaccuracies, which can become a major problem at the scale of existing models, especially when combined with the block updates discussed in the previous step. Specifically, it can occur that the matrix $\mathbf{H}_F^{-1}$ becomes indefinite, which we notice can cause the algorithm to aggressively update the remaining weights in incorrect directions, resulting in an arbitrarily-bad quantization of the corresponding layer. In practice, we observed that the probability of this happening increases with model size: concretely, it almost certainly occurs for at least a few layers on models that are larger than a few billion parameters. The main issue appears to be the repeated applications of Equation (5), which accumulate various numerical errors, especially through the additional matrix inversion.

For smaller models, applying dampening, that is adding a small constant $\lambda$ (we always choose 1% of the average diagonal value) to the diagonal elements of $\mathbf{H}$ appears to be sufficient to avoid numerical issues. However, larger models require a more robust and general approach.

To address this, we begin by noting that the only information required from $\mathbf{H}_{F_q}^{-1}$, where $F_q$ denotes the set of unquantized weights when quantizing weight $q$, is row $q$, or more precisely, the elements in this row starting with the diagonal. The consequence is that we could precompute all of these rows using a more numerically-stable method without any significant increase in memory consumption. Indeed, the row removal via (3) for our symmetric $\mathbf{H}^{-1}$ essentially corresponds to taking a Cholesky decomposition, except for the minor difference that the latter divides row $q$ by $([\mathbf{H}_{F_q}^{-1}]_{qq})^{1/2}$. Hence, we can leverage state-of-the-art Cholesky kernels to compute all information we will need from $\mathbf{H}^{-1}$ upfront. In combination with mild dampening, the resulting method is robust enough to execute on huge models without issues. As a bonus, using a well-optimized Cholesky kernel also yields further speedup. We detail all small changes necessary for the Cholesky version of the algorithm next.

**The Full Algorithm.** Finally, we present the full pseudocode for OPTQ in Algorithm 1, including the optimizations discussed above.

---

**Algorithm 1** Quantize $\mathbf{W}$ given inverse Hessian $\mathbf{H}^{-1} = (2\mathbf{X}\mathbf{X}^\top + \lambda\mathbf{I})^{-1}$ and blocksize $B$.

---

$\mathbf{Q} \leftarrow \mathbf{0}_{d_{\text{row}} \times d_{\text{col}}}$      // quantized output
$\mathbf{E} \leftarrow \mathbf{0}_{d_{\text{row}} \times B}$      // block quantization errors
$\mathbf{H}^{-1} \leftarrow \text{Cholesky}(\mathbf{H}^{-1})^\top$      // Hessian inverse information
**for** $i = 0, B, 2B, \ldots$ **do**
    **for** $j = i, \ldots, i + B - 1$ **do**
        $\mathbf{Q}_{:,j} \leftarrow \text{quant}(\mathbf{W}_{:,j})$      // quantize column
        $\mathbf{E}_{:,j-i} \leftarrow (\mathbf{W}_{:,j} - \mathbf{Q}_{:,j}) / [\mathbf{H}^{-1}]_{jj}$      // quantization error
        $\mathbf{W}_{:,j:(i+B)} \leftarrow \mathbf{W}_{:,j:(i+B)} - \mathbf{E}_{:,j-i} \cdot \mathbf{H}^{-1}_{j,j:(i+B)}$      // update weights in block
    **end for**
    $\mathbf{W}_{:,(i+B):} \leftarrow \mathbf{W}_{:,(i+B):} - \mathbf{E} \cdot \mathbf{H}^{-1}_{i:(i+B),(i+B):}$      // update all remaining weights
**end for**

---

## 5  EXPERIMENTAL VALIDATION

**Overview.** We begin our experiments by validating the accuracy of OPTQ relative to other accurate-but-expensive quantizers, on smaller models, for which these methods provide reasonable runtimes. Next, we examine OPTQ's runtime scaling for very large models. Then, we present 3- and 4-bit quantization results for the entire BLOOM and OPT model families, evaluated via perplexity on challenging language generation tasks. In addition, we show that our method is also stable for 2-bit quantization when the granularity is reduced to small blocks of consecutive weights. To complement this perplexity analysis, we also evaluate the resulting quantized models on a series of standard zero-shot tasks. Finally, we focus on the two largest (and interesting) openly-available models, Bloom-176B and OPT-175B, where we perform a detailed evaluation on several tasks. For these models, we also present practical improvements, namely reducing the number of GPUs required for inference as well as end-to-end speedups for generative tasks.

**Setup.** We implemented OPTQ in PyTorch (Paszke et al., 2019) and worked with the HuggingFace integrations of the BLOOM (Laurençon et al., 2022) and OPT (Zhang et al., 2022) model families. We quantized all models (including the 175 billion parameter variants) *using a single NVIDIA A100 GPU* with 80GB of memory. Our entire OPTQ calibration data consists of 128 random 2048 token segments from the C4 dataset (Raffel et al., 2020), i.e., excerpts from randomly crawled websites, which represents generic text data. We emphasize that this means that OPTQ does not see any task-specific data, and our results thus remain actually "zero-shot". We perform standard uniform per-row asymmetric quantization on the min-max grid, similar to Dettmers et al. (2022). Additional evaluation details can be found in Appendix A.2.1.

To ensure that the entire compression procedure can be performed with significantly less GPU memory than what would be required to run the full precision model, some care must be taken. Specifically, we always load one Transformer block, consisting of 6 layers, at a time into GPU memory and then accumulate the layer-Hessians and perform quantization. Finally, the current block inputs are sent through the fully quantized block again to produce the new inputs for the quantization of the next block. Hence, the quantization process operates not on the layer inputs in the full precision model but on the actual layer inputs in the already partially quantized one. We find that this brings noticeable improvements at negligible extra cost.

**Baselines.** Our primary baseline, denoted by RTN, consists of rounding all weights to the nearest quantized value on exactly the same asymmetric per-row grid that is also used for OPTQ, meaning that it corresponds precisely to the state-of-the-art weight quantization of LLM.int8(). This is currently the method of choice in all works on quantization of very large language models (Dettmers et al., 2022; Yao et al., 2022; Park et al., 2022): its runtime scales well to networks with many billions of parameters, as it simply performs direct rounding. As we will also discuss further, more accurate methods, such as AdaRound (Nagel et al., 2020) or BRECQ (Li et al., 2021), are currently too slow for models with many billions of parameters, the main focus of this work. Nevertheless, we also show that OPTQ is competitive with such methods for small models, while scaling to huge ones like OPT-175B as well.

**Quantizing Small Models.** As a first ablation study, we compare OPTQ's performance relative to state-of-the-art post-training quantization (PTQ) methods, on ResNet18 and ResNet50, which are standard PTQ benchmarks, in the same setup as (Frantar et al., 2022). As can be seen in Table 1, OPTQ performs on par at 4-bit, and slightly worse than the most accurate methods at 3-bit. At the same time, it significantly outperforms AdaQuant, the fastest amongst prior PTQ methods. Further, we compare against the full greedy OBQ method on two smaller language models: BERT-base (Devlin et al., 2019) and OPT-125M. The results are shown in Appendix Table 8. At 4 bits, both methods perform similarly, and for 3 bits, OPTQ surprisingly performs slightly better. We suspect that this is because some of the additional heuristics used by OBQ, such as early outlier rounding, might require careful adjustments for optimal performance on non-vision models. Overall, OPTQ appears to be competitive with state-of-the-art post-training methods for smaller models, while taking only $< 1$ minute rather than $\approx 1$ hour. This enables scaling to much larger models.

**Runtime.** Next we measure the full model quantization time (on a single NVIDIA A100 GPU) via OPTQ; the results are shown in Table 2. As can be seen, OPTQ quantizes 1-3 billion parameter models in a matter of minutes and 175B ones in a few hours. For reference, the straight-through based method ZeroQuant-LKD (Yao et al., 2022) reports a 3 hour runtime (on the same hardware) for a 1.3B model, which would linearly extrapolate to several hundred hours (a few weeks) for 175B

| Method | RN18 – 69.76 % | | RN50 – 76.13% | |
|---|---|---|---|---|
| | 4bit | 3bit | 4bit | 3bit |
| AdaRound | 69.34 | 68.37 | 75.84 | 75.14 |
| AdaQuant | 68.12 | 59.21 | 74.68 | 64.98 |
| BRECQ | 69.37 | 68.47 | 75.88 | 75.32 |
| OBQ | 69.56 | 68.69 | 75.72 | 75.24 |
| OPTQ | 69.37 | 67.88 | 75.71 | 74.87 |

Table 1: Comparison with state-of-the-art post-training methods for vision models.

| OPT | 13B | 30B | 66B | 175B |
|---|---|---|---|---|
| Runtime | 20.9m | 44.9m | 1.6h | 4.2h |
| BLOOM | 1.7B | 3B | 7.1B | 176B |
| Runtime | 2.9m | 5.2m | 10.0m | 3.8h |

Table 2: OPTQ runtime for full quantization of the 4 largest OPT and BLOOM models.

models. Adaptive rounding-based methods typically employ a lot more SGD steps and would thus be even more expensive (Nagel et al., 2020; Li et al., 2021).

**Language Generation.** We begin our large-scale study by compressing the entire OPT and BLOOM model families to 3- and 4-bit. We then evaluate those models on several language tasks including WikiText2 (Merity et al., 2016) (see Figure 1 as well as Tables 3 and 4), Penn Treebank (PTB) (Marcus et al., 1994) and C4 (Raffel et al., 2020) (both in Appendix A.3). We focus on these perplexity-based tasks, as they are known to be particularly sensitive to model quantization (Yao et al., 2022). On OPT models, OPTQ clearly outperforms RTN, by significant margins. For example, OPTQ loses only 0.03 perplexity at 4-bit on the 175B model, while RTN drops 2.2 points, performing worse than the $10\times$ smaller full-precision 13B model. At 3-bit, RTN collapses completely, while OPTQ can still maintain reasonable perplexity, in particular for larger models. BLOOM shows a similar pattern: the gaps between methods are however usually a bit smaller, indicating that this model family might be easier to quantize. One interesting trend (see also Figure 1) is that larger models generally (with the exception of OPT-66B[2]) appear easier to quantize. This is good news for practical applications, as these are the cases where compression is also the most necessary.

| OPT | Bits | 125M | 350M | 1.3B | 2.7B | 6.7B | 13B | 30B | 66B | 175B |
|---|---|---|---|---|---|---|---|---|---|---|
| full | 16 | 27.65 | 22.00 | 14.63 | 12.47 | 10.86 | 10.13 | 9.56 | 9.34 | 8.34 |
| RTN | 4 | 37.28 | 25.94 | 48.17 | 16.92 | 12.10 | 11.32 | 10.98 | 110 | 10.54 |
| OPTQ | 4 | **31.12** | **24.24** | **15.47** | **12.87** | **11.39** | **10.31** | **9.63** | **9.55** | **8.37** |
| RTN | 3 | 1.3e3 | 64.57 | 1.3e4 | 1.6e4 | 5.8e3 | 3.4e3 | 1.6e3 | 6.1e3 | 7.3e3 |
| OPTQ | 3 | **53.85** | **33.79** | **20.97** | **16.88** | **14.86** | **11.61** | **10.27** | **14.16** | **8.68** |

Table 3: OPT perplexity results on WikiText2.

| BLOOM | Bits | 560M | 1.1B | 1.7B | 3B | 7.1B | 176B |
|---|---|---|---|---|---|---|---|
| full | 16 | 22.42 | 17.69 | 15.39 | 13.48 | 11.37 | 8.11 |
| RTN | 4 | 25.90 | 22.00 | 16.97 | 14.76 | 12.10 | 8.37 |
| OPTQ | 4 | **24.03** | **19.05** | **16.48** | **14.20** | **11.73** | **8.21** |
| RTN | 3 | 57.08 | 50.19 | 63.59 | 39.36 | 17.38 | 571 |
| OPTQ | 3 | **32.31** | **25.08** | **21.11** | **17.40** | **13.47** | **8.64** |

Table 4: BLOOM perplexity results for WikiText2.

**175 Billion Parameter Models.** We now examine BLOOM-176B and OPT-175B, the largest dense openly-available models. Table 5 summarizes results across Wikitext-2, PTB, C4. We observe that, at 4 bits, OPTQ models reach only $\leq 0.25$ lower perplexity than the full-precision versions, with a large gap to RTN results on OPT-175B. At 3-bit, RTN collapses, while OPTQ is still able to maintain good performance on most tasks, losing only $0.3 - 0.6$ points for more than $5\times$ compression. We note that OPTQ's accuracy can be further improved via finer-granularity grouping (Park et al., 2022): group-size 1024 ($\approx 0.02$ extra bits) improves perplexities by about 0.2 on average and group-size 128 ($\approx 0.15$ extra bits) by another 0.1, which is only $0.1 - 0.3$ off from the uncompressed accuracy.

---

[2]Upon closer inspection of the OPT-66B model, it appears that this is correlated with the fact that this trained model has a significant fraction of dead units in the early layers, which may make it harder to compress.

We note that grouping interacts very well with OPTQ, as the group parameters can be determined during the quantization process of each layer, always using the most current updated weights.

| Method | Bits | OPT-175B | | | | BLOOM-176B | | | |
|---|---|---|---|---|---|---|---|---|---|
| | | Wiki2 | PTB | C4 | LAMB. ↑ | Wiki2 | PTB | C4 | LAMB. ↑ |
| Baseline | 16 | 8.34 | 12.01 | 10.13 | 75.59 | 8.11 | 14.59 | 11.71 | 67.40 |
| RTN | 4 | 10.54 | 14.22 | 11.61 | 71.34 | 8.37 | 15.00 | 12.04 | 66.70 |
| OPTQ | 4 | **8.37** | **12.26** | **10.28** | **76.80** | **8.21** | **14.75** | **11.81** | **67.71** |
| RTN | 3 | 7.3e3 | 8.0e3 | 4.6e3 | 0 | 571. | 107. | 598. | 0.17 |
| OPTQ | 3 | **8.68** | **12.68** | **10.67** | **76.19** | **8.64** | **15.57** | **12.27** | **65.10** |
| OPTQ | 3/g1024 | 8.45 | 12.48 | 10.47 | 77.39 | 8.35 | 15.01 | 11.98 | 67.47 |
| OPTQ | 3/g128 | 8.45 | 12.37 | 10.36 | 76.42 | 8.26 | 14.89 | 11.85 | 67.86 |

Table 5: Results summary for OPT-175B and BLOOM-176B. "g1024" and "g128" denote results with groupings of size 1024 and 128, respectively.

**Practical Speedups.** Finally, we study practical applications. As an interesting use-case, we focus on the OPT-175B model: quantized to 3 bits, this model takes approximately 63GB of memory, including the embeddings and the output layer, which are kept in full FP16 precision. Additionally, storing the complete history of keys and values for all layers, a common optimization for generation tasks, consumes another $\approx$ 9GB for the maximum of 2048 tokens. Hence, we can actually fit the entire quantized model into a single 80GB A100 GPU, which can be executed by dynamically dequantizing layers as they are required during inference (the model would not fully fit using 4 bits). For reference, standard FP16 execution requires 5x80GB GPUs, and the state-of-the-art 8bit LLM.int8() quantizer (Dettmers et al., 2022) requires 3 such GPUs.

Next, we consider language generation, one of the most appealing applications of these models, with the goal of latency reduction. Unlike LLM.int8(), which reduces memory costs but has the same runtime as the FP16 baseline, we show that our quantized models can achieve significant speedups for this application. For language generation, the model processes and outputs one token at-a-time, which for OPT-175B can easily take a few 100s of milliseconds per token. Increasing the speed at which the user receives generated results is challenging, as compute is dominated by matrix-vector products. Unlike matrix-matrix products, these are primarily limited by memory bandwidth. We address this problem by developing a quantized-matrix full-precision-vector product kernel which performs a matrix vector product by dynamically dequantizing weights when needed. Most notably, this does *not* require any activation quantization. While dequantization consumes extra compute, the kernel has to access a lot less memory, leading to significant speedups, as shown in Table 6. We note that almost all of the speedup is due to our kernels, as communication costs are negligible in our standard HuggingFace-accelerate-like setting (see Appendix A.2.2 for details).

| GPU | FP16 | 3bit | Speedup | GPU reduction |
|---|---|---|---|---|
| A6000 − 48GB | 589ms | 130ms | 4.53× | 8 → 2 |
| A100 − 80GB | 230ms | 71ms | 3.24× | 5 → 1 |

Table 6: Average per-token latency (batch size 1) when generating sequences of length 128.

For example, using our kernels, the 3-bit OPT-175B model obtained via OPTQ running on a single A100 is about $3.25\times$ faster than the FP16 version (running on 5 GPUs) in terms of average time per token. More accessible GPUs, such as the NVIDIA A6000, have much lower memory bandwidth, so this strategy is even more effective: executing the 3-bit OPT-175B model on 2x A6000 GPUs reduces latency from 589 milliseconds for FP16 inference (on 8 GPUs) to 130 milliseconds, a $4.5\times$ latency reduction.

**Zero-Shot Tasks.** While our focus is on language generation, we also evaluate the performance of quantized models on some popular zero-shot tasks, namely LAMBADA (Paperno et al., 2016), ARC (Easy and Challenge) (Boratko et al., 2018) and PIQA (Tata & Patel, 2003). Figure 3 visualizes model performance on LAMBADA (and see also "Lamb." results in Table 5). We observe similar behavior as before: the outliers are that 1) quantization appears "easier" across the whole spectrum of models at 4-bit, where even RTN performs relatively well, and 2) at 3-bit, RTN breaks down, while OPTQ still provides good accuracy. We provide additional results in Appendix A.4.

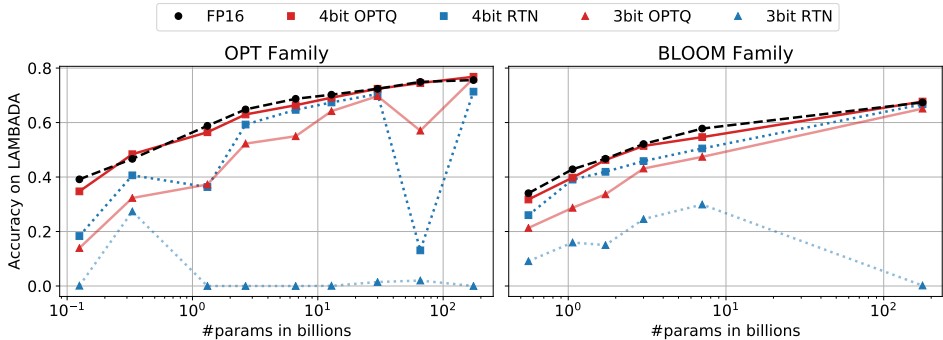

Figure 3: The accuracy of OPT and BLOOM models post-OPTQ, measured on LAMBADA.

**Additional Tricks.** While our experiments so far have focused exclusively on vanilla row-wise quantization, we want to emphasize that OPTQ is *compatible with essentially any choice of quantization grid*. For example, it is easily combined with standard *grouping* (Alistarh et al., 2017; Park et al., 2022), i.e. applying independent quantization to groups of $g$ consecutive weights. As shown in the last rows of Table 5, this can bring noticeable extra accuracy for the largest models at 3-bit. Further, as visualized in Figure 4, it significantly reduces the accuracy losses for medium sized models at 4-bit precision.

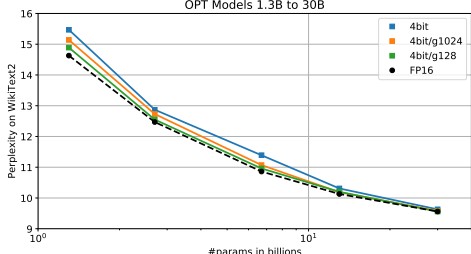

| Model | FP16 | g128 | g64 | g32 | 3-bit |
|---|---|---|---|---|---|
| OPT-175B | 8.34 | 9.58 | 9.18 | 8.94 | 8.68 |
| BLOOM | 8.11 | 9.55 | 9.17 | 8.83 | 8.64 |

Table 7: 2-bit OPTQ quantization results with varying group-sizes; perplexity on WikiText2.

Figure 4: OPTQ at 4-bit with different group-sizes on medium sized OPT models.

**Extreme Quantization.** Lastly, grouping also makes it possible to achieve reasonable performance for extreme quantization, to around 2-bits per component on average. Table 7 shows results on WikiText2 when quantizing the biggest models to 2-bit with varying group-sizes. At $\approx$ 2.2 bit (group-size 128; using FP16 scale and 2-bit zero point per group) the perplexity increase is already less than 1.5 points, while dropping to 0.6 - 0.7 at $\approx$ 2.6 bit (group-size 32), which is only slightly worse than vanilla 3-bit and might be interesting for practical kernel implementations. Further, if we reduce group size to 8, we can apply *ternary* (-1, 0, +1) quantization, which achieves 9.20 WikiText2 PPL on OPT-175B, a less than 1 point drop. While this leads to worse compression on average relative to the 2-bit numbers above, this pattern could be efficiently implemented on custom hardware such as FPGAs. In summary, these results are an encouraging first step towards pushing highly-accurate *one-shot* compression of very large language models, even lower than 3 bits per value on average.

## 6  SUMMARY AND LIMITATIONS

We have presented OPTQ, an approximate second-order method for quantizing truly large language models. OPTQ can accurately compress some of the largest publicly-available models down to 3 and 4 bits, which leads to significant usability improvements, and to end-to-end speedups, at low accuracy loss. We hope that our method will make these models accessible to more researchers and practitioners. At the same time, we emphasize some significant limitations: On the technical side, our method obtains speedups from reduced memory movement, and does not lead to computational reductions. In addition, our study focuses on generative tasks, and does not consider activation quantization. These are natural directions for future work, and we believe this can be achieved with carefully-designed GPU kernels and existing techniques (Yao et al., 2022; Wu et al., 2022).

ACKNOWLEDGMENTS

Elias Frantar and Dan Alistarh gratefully acknowledge funding from the European Research Council (ERC) under the European Union's Horizon 2020 programme (grant agreement No. 805223 ScaleML), as well as experimental support from Eldar Kurtic, and from the IST Austria IT department, in particular Stefano Elefante, Andrei Hornoiu, and Alois Schloegl. The work of Saleh Ashkboos and Torsten Hoefler was supported by the PASC DaCeMI project, received EuroHPC-JU funding under grant MAELSTROM, No. 955513. We thank the Swiss National Supercomputing Center (CSCS) for supporting us with compute infrastructure.

## 7   ETHICS STATEMENT

Our work introduces a general method for compressing large language models (LLMs) via quantization, with little-to-no accuracy loss in terms of standard accuracy metrics such as perplexity. Our method is task-agnostic, as it only uses a tiny amount of randomly-chosen data for calibration. We therefore do not foresee any significant ethical implications arising directly from the technical details of our method. However, one possible consideration is that our study focused on "leading accuracy" metrics that are standard in the literature, such as perplexity, which is essentially standard in the literature (Dettmers et al., 2022; Yao et al., 2022). We believe a thorough study of the impact of compression upon secondary measures, and in particular bias effects (Bender et al., 2021) is warranted, and may be rendered easier through our work. At the same time, our work makes inference on extremely large language models more accessible, for better or for worse. We believe that, in time, such tools will become much easier to use and deploy, making the need to understand their power and limitations even more stringent.

## 8   REPRODUCIBILITY STATEMENT

In the Supplementary Materials, we provide code to reproduce all experiments in this paper. More specifically, this includes:

- Compressing all models from the OPT and BLOOM model families to 2/3/4 bits.
- Evaluating perplexity of the quantized models.
- Our 3-bit CUDA kernel together with compressed inference benchmarking features.
- Code for the ZeroShot experiments.
- A README file providing sample commands and information on how to run all scripts.

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

# A APPENDIX

## A.1 ADDITIONAL COMPARISON WITH OBQ

We now provide an additional comparison between OPTQ and OBQ on BERT-base/SQuAD Rajpurkar et al. (2016) and OPT-125M/WikiText2, which is one of the largest models to which OBQ can be reasonably applied.

| Method | BERT-base 88.53 F1 ↑ | | OPT-125M 27.66 PPL ↓ | |
|--------|------|------|------|------|
| | 4bit | 3bit | 4bit | 3bit |
| OBQ | 88.23 | 85.29 | 32.52 | 69.32 |
| OPTQ | 88.18 | 86.02 | 31.12 | 53.85 |

Table 8: Comparison of OPTQ relative to OBQ on BERT-base/SQuAD and OPT-125M/WikiText2.

## A.2 EXPERIMENT DETAILS

This section provides additional details about our experiment setup, in particular regarding the model evaluation and the setup of our timing experiments.

### A.2.1 EVALUATION

For language generation experiments, we calculate the perplexity, in standard fashion like Radford et al. (2019), as follows: First, the entire validation set is concatenated using two linebreaks as separators and encoded using the default HuggingFace tokenizer of each model. Next, the sequence is split into non-overlapping segments of width 2048, the full context size of our models. These are sent through the model to collect the log-probabilities corresponding to the next token each. Their exponentiated average is the final perplexity we report.

For zero-shot tasks we follow the EleutherAI evaluation harness[3] in terms of data preprocessing and final score calculation. We note that we evaluate all individual samples separately and thus do not apply any padding.

### A.2.2 TIMING EXPERIMENT SETUP

Our timing experiments are performed following the standard HuggingFace/accelerate[4] setup also used by the recent work LLM.int8() (Dettmers et al., 2022). In this setting, the model is split by distributing chunks of consecutive layers across GPUs. Importantly, in this setup the communication costs are minimal, $< 5\%$ of the total runtime even when working with 8 GPUs. This means almost all of the reported speedups are due to our quantized-matrix full-precision vector product kernels. We emphasize that the only difference between the FP16 baseline and our quantized models are the kernels used to perform the underlying matrix-vector products.

This means all overheads due to HuggingFace, attention or non-quantized operations like residuals or LayerNorms are exactly the same. Consequently, our quantized models should benefit from more advanced distribution strategies (Zheng et al., 2022) or more efficient attention kernels (Dao et al., 2022) just as much as our baseline.

In general, our kernels target generative inference in the low batch-size setting (for simplicity, we consider only batchsize 1) where the underlying (close to) matrix-vector products are memory-bound. For non-generative and large-batch applications, operations may be compute- rather than memory-bound and our kernels thus not directly applicable. Instead, one could simply decompress the matrix before performing the corresponding matrix-matrix calculations: this takes $< 1.5$ms on an A100 and $< 3$ms on an A6000 compared to 76ms/365ms for the subsequent OPT-175B FC2 layer computation with batchsize $16 \times 1024$ tokens. Hence, for such applications our methods significantly reduce the required number of GPUs at very little computational overhead. This is similar to recent work (Dettmers et al., 2022), but we achieve a $2.5\times$ higher compression rate.

---

[3]https://github.com/EleutherAI/lm-evaluation-harness
[4]https://huggingface.co/docs/accelerate/index

## A.3 ADDITIONAL LANGUAGE GENERATION RESULTS

Tables 9, 10, 11 and 12 show additional results for language generation tasks.

| OPT | Bits | 125M | 350M | 1.3B | 2.7B | 6.7B | 13B | 30B | 66B | 175B |
|------|------|-------|-------|-------|-------|-------|-------|-------|--------|-------|
| full | 16 | 38.99 | 31.08 | 20.29 | 17.97 | 15.77 | 14.52 | 14.04 | 13.36 | 12.01 |
| RTN | 4 | 53.89 | 36.79 | 57.30 | 31.05 | 18.84 | 16.51 | 15.40 | 225.66 | 14.22 |
| OPTQ | 4 | **45.17** | **34.52** | **21.85** | **19.14** | **16.56** | **14.94** | **14.26** | **13.81** | **12.26** |
| RTN | 3 | 1.4e3 | 88.04 | 1.3e4 | 1.4e4 | 5.7e3 | 2.8e3 | 1.2e3 | 5.0e3 | 8.0e3 |
| OPTQ | 3 | **73.19** | **47.08** | **32.10** | **24.81** | **21.88** | **16.68** | **15.36** | **28.12** | **12.86** |

Table 9: OPT perplexity results on PTB.

| BLOOM | Bits | 560M | 1.1B | 1.7B | 3B | 7.1B | 176B |
|--------|------|-------|-------|-------|-------|-------|-------|
| full | 16 | 43.69 | 57.96 | 30.00 | 25.34 | 20.83 | 14.59 |
| RTN | 4 | 51.10 | 66.85 | 33.58 | 27.68 | 22.42 | 15.00 |
| OPTQ | 4 | **46.97** | **62.47** | **31.84** | **26.49** | **21.67** | **14.75** |
| RTN | 3 | 126. | 185. | 106. | 66.78 | 35.04 | 107. |
| OPTQ | 3 | **70.35** | **87.04** | **46.11** | **34.02** | **26.14** | **15.57** |

Table 10: BLOOM perplexity results for PTB.

| OPT | Bits | 125M | 350M | 1.3B | 2.7B | 6.7B | 13B | 30B | 66B | 175B |
|------|------|-------|-------|-------|-------|-------|-------|-------|-------|-------|
| full | 16 | 26.56 | 22.59 | 16.07 | 14.34 | 12.71 | 12.06 | 11.44 | 10.99 | 10.13 |
| RTN | 4 | 33.91 | 26.21 | 24.51 | 18.43 | 14.36 | 13.36 | 13.46 | 309. | 11.61 |
| OPTQ | 4 | **29.22** | **24.63** | **16.97** | **15.00** | **13.18** | **12.26** | **11.57** | **11.23** | **10.28** |
| RTN | 3 | 834 | 55.49 | 5.2e3 | 1.1e4 | 5.3e3 | 3.1e3 | 1.4e3 | 3.5e3 | 4.6e3 |
| OPTQ | 3 | **42.41** | **31.33** | **21.63** | **18.17** | **17.14** | **13.34** | **12.23** | **14.59** | **10.67** |

Table 11: OPT perplexity results on C4. We note that the calibration data used by OPTQ is sampled from the C4 training set, this task is thus not fully zero-shot.

| BLOOM | Bits | 560M | 1.1B | 1.7B | 3B | 7.1B | 176B |
|--------|------|-------|-------|-------|-------|-------|-------|
| full | 16 | 26.60 | 22.05 | 19.49 | 17.49 | 15.20 | 11.71 |
| RTN | 4 | 29.89 | 24.44 | 21.26 | 18.76 | 16.06 | 12.04 |
| OPTQ | 4 | **28.00** | **23.25** | **20.55** | **18.10** | **15.60** | **11.81** |
| RTN | 3 | 67.49 | 60.71 | 113. | 80.49 | 22.59 | 598. |
| OPTQ | 3 | **35.78** | **28.83** | **25.34** | **21.25** | **17.67** | **12.27** |

Table 12: BLOOM perplexity results for C4. We note that the calibration data used by OPTQ is sampled from the C4 training set, this task is thus not fully zero-shot.

## A.4 ADDITIONAL ZEROSHOT RESULTS

This section contains additional results for zero-shot tasks.

| OPT | Bits | 125M | 350M | 1.3B | 2.7B | 6.7B | 13B | 30B | 66B | 175B |
|-----|------|------|------|------|------|------|------|------|------|------|
| full | 16 | 39.16 | 46.67 | 58.80 | 64.82 | 68.72 | 70.23 | 72.39 | 74.93 | 75.59 |
| RTN | 4 | 18.34 | 40.62 | 36.31 | 59.27 | 64.66 | 67.38 | 70.48 | 13.08 | 71.34 |
| OPTQ | 4 | **34.74** | **48.38** | **56.45** | **62.97** | **66.37** | **69.12** | **72.40** | **74.50** | **76.80** |
| RTN | 3 | 0.10 | 27.36 | 0.00 | 0.00 | 0.00 | 0.06 | 1.46 | 2.00 | 0.00 |
| OPTQ | 3 | **13.93** | **32.31** | **37.26** | **52.26** | **54.98** | **64.18** | **69.69** | **57.02** | **76.19** |

Table 13: OPT accuracy on LAMBADA.

| BLOOM | Bits | 560M | 1.1B | 1.7B | 3B | 7.1B | 176B |
|-------|------|------|------|------|-----|------|------|
| full | 16 | 34.06 | 42.85 | 46.71 | 52.12 | 57.79 | 67.40 |
| RTN | 4 | 26.00 | 39.06 | 41.92 | 45.84 | 50.48 | 66.70 |
| OPTQ | 4 | **31.75** | **39.80** | **46.28** | **51.41** | **54.65** | **67.71** |
| RTN | 3 | 9.10 | 15.95 | 15.02 | 24.55 | 29.90 | 0.17 |
| OPTQ | 3 | **21.31** | **28.70** | **33.65** | **43.12** | **47.41** | **65.10** |

Table 14: BLOOM accuracy on LAMBADA.

| OPT | Bits | 125M | 350M | 1.3B | 2.7B | 6.7B | 13B | 30B | 66B | 175B |
|-----|------|------|------|------|------|------|------|------|------|------|
| full | 16 | 62.02 | 64.74 | 72.36 | 74.81 | 76.39 | 76.88 | 78.18 | 79.76 | 81.07 |
| RTN | 4 | **61.43** | 63.44 | 67.63 | 73.72 | **76.44** | 76.01 | 77.26 | 60.07 | 78.23 |
| OPTQ | 4 | 61.26 | **63.71** | **70.73** | **73.99** | 76.28 | **76.61** | **79.00** | **79.33** | **81.00** |
| RTN | 3 | 56.09 | 60.61 | 52.77 | 51.90 | 50.49 | 52.99 | 56.37 | 50.87 | 51.25 |
| OPTQ | 3 | **59.25** | **61.32** | **68.34** | **71.38** | **73.29** | **75.24** | **77.58** | **71.27** | **80.03** |

Table 15: OPT accuracy on PIQA.

| BLOOM | Bits | 560M | 1.1B | 1.7B | 3B | 7.1B | 176B |
|-------|------|------|------|------|-----|------|------|
| full | 16 | 65.07 | 67.14 | 69.97 | 70.51 | 73.72 | 79.16 |
| RTN | 4 | 63.11 | 65.29 | 67.74 | **69.86** | 72.69 | **79.00** |
| OPTQ | 4 | **64.31** | **66.05** | **68.77** | 69.42 | **72.96** | **79.00** |
| RTN | 3 | 58.60 | 60.80 | 60.88 | 66.28 | 69.70 | 53.32 |
| OPTQ | 3 | **61.62** | **62.62** | **65.18** | **68.34** | **70.95** | **77.70** |

Table 16: BLOOM accuracy on PIQA.

| OPT | Bits | 125M | 350M | 1.3B | 2.7B | 6.7B | 13B | 30B | 66B | 175B |
|-----|------|------|------|------|------|------|------|------|------|------|
| full | 16 | 39.69 | 40.36 | 50.93 | 54.34 | 60.14 | 61.83 | 65.40 | 67.26 | 71.04 |
| RTN | 4 | 36.32 | **38.55** | 49.20 | 52.90 | 57.68 | 61.31 | 61.11 | 40.66 | 63.93 |
| OPTQ | 4 | **39.02** | 37.92 | **59.97** | **53.11** | **59.72** | **61.32** | **65.11** | **65.35** | **68.69** |
| RTN | 3 | 30.43 | 36.07 | 27.97 | 26.05 | 25.04 | 30.60 | 34.22 | 25.84 | 26.77 |
| OPTQ | 3 | **36.15** | **36.91** | **46.17** | **48.19** | **53.41** | **56.82** | **59.72** | **52.44** | **65.36** |

Table 17: OPT accuracy on ARC-easy.

| BLOOM | Bits | 560M | 1.1B | 1.7B | 3B | 7.1B | 176B |
|---|---|---|---|---|---|---|---|
| full | 16 | 41.71 | 45.41 | 48.11 | 53.24 | 57.37 | 67.47 |
| RTN | 4 | 39.40 | 42.51 | **44.70** | 51.35 | **56.14** | 66.33 |
| OPTQ | 4 | **40.24** | **44.49** | 44.49 | **52.82** | **56.14** | **67.42** |
| RTN | 3 | **45.44** | **46.87** | 37.58 | 45.08 | 48.61 | 28.87 |
| OPTQ | 3 | 39.14 | 41.79 | **42.85** | **46.63** | **51.56** | **62.84** |

Table 18: BLOOM accuracy on ARC-easy.

| OPT | Bits | 125M | 350M | 1.3B | 2.7B | 6.7B | 13B | 30B | 66B | 175B |
|---|---|---|---|---|---|---|---|---|---|---|
| full | 16 | 22.87 | 24.06 | 29.44 | 31.31 | 34.56 | 35.75 | 38.14 | 40.02 | 43.94 |
| RTN | 4 | 22.44 | 23.81 | 24.91 | 29.18 | 32.59 | **35.24** | 35.41 | 22.87 | 37.71 |
| OPTQ | 4 | **22.95** | **24.83** | **28.24** | **30.12** | **33.70** | 34.90 | **37.80** | **39.16** | **42.75** |
| RTN | 3 | 21.76 | 22.18 | 23.55 | 25.43 | 25.85 | 23.81 | 19.97 | 25.77 | 23.81 |
| OPTQ | 3 | **22.53** | **25.09** | **27.65** | **27.82** | **31.91** | **33.02** | **35.84** | **31.66** | **41.04** |

Table 19: OPT accuracy on ARC-challenge.

| BLOOM | Bits | 560M | 1.1B | 1.7B | 3B | 7.1B | 176B |
|---|---|---|---|---|---|---|---|
| full | 16 | 24.15 | 25.68 | 26.79 | 30.55 | 33.45 | 44.97 |
| RTN | 4 | **23.89** | 23.34 | **26.45** | **29.52** | 32.17 | 43.17 |
| OPTQ | 4 | 23.46 | **25.51** | 25.94 | 28.92 | **32.25** | **44.20** |
| RTN | 3 | 21.67 | 22.86 | 23.29 | 27.13 | **31.31** | 24.74 |
| OPTQ | 3 | **23.21** | **24.06** | **24.91** | **28.58** | 30.97 | **40.70** |

Table 20: BLOOM accuracy on ARC-challenge.

| OPT | Bits | 125M | 350M | 1.3B | 2.7B | 6.7B | 13B | 30B | 66B | 175B |
|---|---|---|---|---|---|---|---|---|---|---|
| full | 16 | 59.96 | 63.21 | 70.78 | 71.74 | 74.60 | 76.64 | 77.28 | 77.34 | 79.82 |
| RTN | 4 | **60.02** | 63.08 | 59.13 | **70.78** | 73.65 | 74.47 | 75.37 | 51.24 | 78.04 |
| OPTQ | 4 | 59.58 | **63.46** | **69.64** | 70.46 | **73.90** | **76.19** | **77.08** | **77.15** | **80.08** |
| RTN | 3 | 49.65 | 56.78 | 47.61 | 46.98 | 48.12 | 49.20 | 49.84 | 48.19 | 46.47 |
| OPTQ | 3 | **57.03** | **60.15** | **65.25** | **68.43** | **70.97** | **73.07** | **75.68** | **71.23** | **78.04** |

Table 21: OPT accuracy on StoryCloze.

| BLOOM | Bits | 560M | 1.1B | 1.7B | 3B | 7.1B | 176B |
|---|---|---|---|---|---|---|---|
| full | 16 | 61.94 | 63.27 | 65.44 | 67.79 | 71.99 | 76.89 |
| RTN | 4 | 60.15 | 60.66 | 62.95 | 67.09 | 70.72 | 76.00 |
| OPTQ | 4 | **61.17** | **62.32** | **64.48** | **67.22** | **71.36** | **76.32** |
| RTN | 3 | 54.87 | 56.08 | 55.79 | 59.83 | 66.20 | 48.50 |
| OPTQ | 3 | **57.80** | **59.77** | **61.81** | **63.97** | **69.26** | **75.37** |

Table 22: BLOOM accuracy on StoryCloze.

