# OpenReview forum: "OPTQ: Accurate Quantization for Generative Pre-trained Transformers"
_ICLR.cc/2023/Conference — ICLR 2023 poster_

### Official Review · Reviewer_X8QT · 2022-10-23

**Confidence:** 5
**Correctness:** 3
**Technical Novelty And Significance:** 2
**Empirical Novelty And Significance:** 2
**Recommendation:** 5

**Clarity, Quality, Novelty And Reproducibility:**

Section 3.2 is hard to understand without reading previous works, which makes it difficult to follow Section 3.3. It seems that Section 3 should be revised for better understanding. As the GPTQ algorithm is based on OBQ, the novelty of GPTQ seems to be somewhat incremental.

**Strength And Weaknesses:**

-Strengths:
1. This paper well defined the problems that occur in the generation task of the GPT model, and solved it with weight only quantization and FP16 activation.
2. The paper proposes a new weight quantization method named GPTQ, which can quantize the 175B model into 3-bit or 4-bit in a short time with only inside a single GPU.
3. Training and kernel code provided.

-Weaknesses:
1. The presented methodology is marginally improved from the existing method (OBQ).
2. Due to computation time, the authors did not compare the performance with the latest PTQ methodologies. It is required to compare GPTQ with BRECQ for 125M ~ 6.7B models since BRECQ always outperforms GPTQ as seen in Table 1.
3. Based on the code in the supplementary material, when performing layer-wise quantization, the input with all preceding layers quantized is solely utilized. The input with all preceding layers kept in full precision is not exploited at all. Noting that the existing works such as [1] AdaRound and [2] AdaQuant employ both inputs, the proposed method partially use the information about input, which can cause the accumulation of quantization error for deeper networks.
4. The proposed method does not consider the cross-layer dependency at all. As shown in BRECQ, it is far better to consider the cross-layer dependency than not.
5. In $\textbf{H}$ (the Hessian matrix of the reconstruction error, Eq. (1)), which is used for one-shot update of full-precision weights, only the input feature information of the quantized model is used. This may have the following problems.
- Given that taking advantage of $\textbf{H}$ can have nothing to do with the minimization of the final task loss but the weights are already trained towards minimizing the final objective, using $\textbf{H}$ to update weights can deteriorate the performance of layer-wise quantization rather than improve.
- The reason why to update weights trained with the pre-training dataset via $\textbf{H}$ calculated through the calibration set (e.g., C4) is not clear since the pre-training dataset can be irrelevant to the calibration set.

6. PPL measurement is highly dependent on evaluation method. It is necessary to explain how the PPL metric used for evaluation was measured.
- The measured PPL performance of the FP16 model is peculiarly good, considering that all experimental results are based on zero-shot tasks. It would be recommended to supplement by referring to the two links below.
- For fair comparisons, the perplexity should be measured again by referring to the links below.
- https://github.com/huggingface/transformers/tree/main/examples/pytorch/language-modeling
- https://huggingface.co/docs/transformers/perplexity

7. Typo correction required in table2 (66b -> 66B)

**Summary Of The Paper:**

This paper discusses the post-training quantization (PTQ) methodology of a Generative Pre-trained Transformer (GPT) model for a generation task. While formatting the weight to 4 or 3-bit, activation remains as FP. The PTQ methodology is a layer-wise quantization with a small calibration-set, and a part of the weight is quantized using a round-to-nearest (RTN). The remaining FP weights are updated with second-order information (Hessian) of the FP weights through Optimal Brain Quantization (OBQ) in one-shot scheme using the calibration set. Gradually widen the quantization area and quantize the entire weight at the end.

The experiment was performed on the language generation task for the OPT and BLOOM models, and the comparison methodology is FP16 and RTN. Experimental results are shown together with the speedups table as the degradation decreases as the model gradually increases (175B). Additionally, training and kernel code were provided to the supplementary.

**Summary Of The Review:**

The quantization problem of the generative model is realistically solved with weight only quantization and FP activation. The results show a performance close to that of the FP16 in the 175B model, the largest model published. However, the methodology used lacks novelty by utilizing existing ones. In particular, the methodology that repeatedly uses the output of the quantized model from calibration set has not been analyzed how it affects the weight update of the pre-trained model in LLM. Finally, it is necessary to clarify how the PPL metric was measured in the performance evaluation.

Due to such disputable points mentioned in weaknesses and the somewhat incremental novelty of the GPTQ algorithm, I cannot recommend the acceptance of this paper at this moment.

---

> ### Author Response · Authors · 2022-11-08
> **Response to Main Concerns**
>
> Thank you for your detailed comments!
>
> We begin our response by addressing your main concerns, as per your review summary, on the lack of novelty and the perplexity calculation. Afterwards, we will respond to all your other comments as well.
>
> ***
>
> **Improvement over OBQ (item 1):**
>
> We think that there is probably a misunderstanding here on how much more efficient our proposed method actually is: GPTQ improves the theoretical complexity of OBQ by a full factor of $\text{min}(d_\text{row}, d_\text{col})$ and resolves several major practical challenges, leading to an algorithm that is more than 3 orders of magnitude, i.e., 1000x, faster on the very largest models, all while maintaining similar quantization accuracy. Further, GPTQ is not just dramatically more efficient than OBQ but more than 50-100x faster than any other existing post-training method with similar accuracy.
>
> This makes it possible to, for the first time, affordably scale highly accurate post-training quantization to models with 175 billion parameters, enabling 3- and 4-bit quantization at minimal accuracy loss, a more than 2x higher compression relative to existing 8-bit techniques like LLM.int8(). The models we can accurately compress to 4-bit are more than 100x larger than those compressible via existing techniques using similar amounts of compute.
>
> **Perplexity calculation (item 6):**
>
> We agree that a more detailed description of our evaluation process, complementing our code, would be helpful and will add it to the next revision. We calculate perplexity following exactly the process described in your huggingface.co-link, with full 2048 stride for consistency with the GPT evaluation protocol. More precisely, we concatenate the entire validation data into a single sequence using “\n\n” as separator, which is then split into non-overlapping chunks of length 2048, each of which is run through the model to produce the log-likelihoods for each token all of which are averaged and the result ultimately exponentiated to produce the final perplexity.
>
> The reason our FP16 perplexities are quite good is that these tasks are relatively general language modeling tasks and thus not too dissimilar from the actual training distribution. These tasks are also considered by related works on huge model quantization [Yao et al., 2022; Dettmers et al., 2022; Park et al., 2022] who also report similar numbers.  Please note that we also consider quite a few more narrow ZeroShot tasks in the Appendix. We can provide additional validation on this point if the reviewer finds it necessary.

---

> > ### Author Response · Authors · 2022-11-08
> > **Response to Remaining Comments**
> >
> > We now respond to your remaining comments:
> >
> > **Comparison with BRECQ (item 2):**
> >
> > The goal of this work is not to improve the state-of-the-art for models up to a few 100 million parameters but to scale up to models with 100+ billion parameters. We believe that doing both with a single method would be impossible within reasonable time constraints: for instance, BRECQ takes slightly less than one hour to quantize ResNet50, while in the same time GPTQ can compress networks that are 1000x larger. The trend of a well-tuned BRECQ performing slightly better than GPTQ will likely continue on larger models, but at orders of magnitude larger computational cost, rendering a direct comparison not very meaningful.
> >
> > **Full precision model input & cross-layer dependencies (items 3 & 4):**
> >
> > Thank you for these interesting suggestions, which we are planning to investigate in future work.
> >
> > However, we note that maintaining high efficiency when running on extremely large models is very challenging and requires making some extra approximations, some of which will hopefully be relaxed by follow-up work along the lines you suggested.
> >
> > Specifically, on these extremely large networks, overfitting can become a significant problem: just one transformer block of BLOOM-176B contains more than 3 billion parameters, jointly fine tuning it with just 128 samples of 2048 tokens, as we use for GPTQ, will most likely lead to severe overfitting without extremely careful regularization. For illustration, we have observed overfitting issues when attempting to compensate for accumulated errors in similar fashion to the way AdaRound/Quant work. Thus, we decided to stick to the strategy we currently use, which we found to work very reliably.
> >
> > In addition, our experiments demonstrate that our method already achieves strong results in its current form. Specifically, we can go to 3 bits/weight with high accuracy; we will show in the revision that our approach can provide similar perplexities when going to 2 bits per weight, if we quantize in smaller groups. This way, we can achieve good perplexity in one-shot going down to 2.5 bits/weight on average.
> >
> > **Hessian using only input features & calibration data (item 5):**
> >
> > This layer-wise L2 quantization framework is well-established in literature [Nagel et al., 2020; Wang et al., 2020; Hubara et al., 2021; Frantar & Alistarh, 2022], and it is known that layer-wise squared error loss is generally well correlated with final model accuracy; we optimize the same per-layer loss (as virtually all the works above), but using a much more efficient method. Our Hessian calculations and updates are all exact with respect to this layer-wise problem; note that the true Hessian of this squared error does in fact only involve input features, which is also key for the high efficiency of our method. We do agree that attempting to augment the quantization process also with non-input information is a very interesting direction for future work.
> >
> > We sample a calibration dataset from C4, representing generic text data in the form of random text segments crawled from the web, so that GPTQ does not see any data specific to our evaluation task. Calibration data could also come directly from the training set of the respective evaluation task, and we have indeed noticed that this improves results slightly. We chose C4 to keep our method simple and general.
> >
> > **Typo (item 7):**
> >
> > Thank you, we will fix this typo.
> >
> > ***
> >
> > If any questions remain, we are happy to engage in further discussion!

---

> > > ### Comment · Reviewer_X8QT · 2022-11-15
> > > **Response to rebuttal**
> > >
> > > Thank you for trying to address my concerns.
> > >
> > >
> > > - Improvement over OBQ (item 1) & Comparison with BRECQ (item 2):
> > > As pointed out by the reviewer AnMH, it seems to be a marginal improvement over OBQ since Section 3.1 and 3.2 comes from previous works and numerical stability in Section 3.3 is a quite standard technique. So, the novelty of methodology seems to be somewhat incremental.
> > > On top of that, the authors argue that GPTQ is more than 50-100x faster than any other existing post-training method with similar accuracy, but this is totally overstated. First, as I mentioned, BRECQ always outperforms GPTQ as seen in Table 1 (with large margin in 3-bit). In addition, the authors only compare GPTQ with RTN, which is not one of the existing SOTA PTQ methods at all. Therefore, I cannot acknowledge that GPTQ can achieve similar accuracy to any other existing PTQ method at all.
> > >
> > >
> > > - Perplexity calculation (item 6):
> > > In the zero-shot evaluation of the PTB dataset, full-precision OPT-2.7B model shows about 70 ppl when measured using the Huggingface repository code, which is much far from the ppl of full-precision OPT model shown in Table 2 (15.11). Although different results can be yielded depending on how ppl is measured, it would be nice to also show results by following the Huggingface repository code for other researchers. Considering that many NLP researchers use the Huggingface code as a baseline, I believe that it is required to update the score of the paper.
> > >
> > > https://github.com/huggingface/transformers/tree/main/examples/pytorch/language-modeling
> > >
> > >
> > > - Full precision model input & cross-layer dependencies (items 3 & 4):
> > > The authors claim that overfitting can occur when in a similar manner to AdaRound. However, there is no issue about overfitting at all (in papers as well as in my experience) when AdaRound/BRECQ is applied to vision models and BERT. In order to claim that overfitting can occur when in a similar manner to AdaRound, it is totally necessary to explain/illustrate overfitting issues the authors observed in detail.
> > >
> > >
> > > - Hessian using only input features & calibration data (item 5):
> > > As the authors say, the relationship between layer-wise squared error loss and final model accuracy is just the correlation, not the causal relationship. In that sense, it is not still sure why using the hessian matrix of layer-wise squared error loss is beneficial to final (quantized) model accuracy/performance.

---

> > > > ### Author Response · Authors · 2022-11-17
> > > > **Reponse Part 1 (Improvement over OBQ and BRECQ comparison)**
> > > >
> > > >
> > > > We sincerely thank the reviewer for their reply, and address the outstanding questions.
> > > >
> > > > **Improvement over OBQ (item 1) & Comparison with BRECQ (item 2)**
> > > >
> > > > >  As pointed out by the reviewer AnMH, it seems to be a marginal improvement over OBQ since Section 3.1 and 3.2 comes from previous works and numerical stability in Section 3.3 is a quite standard technique. So, the novelty of methodology seems to be somewhat incremental. The authors argue that GPTQ is more than 50-100x faster than any other existing post-training method with similar accuracy, but this is totally overstated. First, as I mentioned, BRECQ always outperforms GPTQ as seen in Table 1 (with large margin in 3-bit). In addition, the authors only compare GPTQ with RTN, which is not one of the existing SOTA PTQ methods at all. Therefore, I cannot acknowledge that GPTQ can achieve similar accuracy to any other existing PTQ method at all.
> > > >
> > > > We wish to clarify the following points regarding contribution and comparisons:
> > > >
> > > > 1. **Our main contribution relative to existing PTQ methods (OBQ, BRECQ, etc) is the fact that GPTQ can work efficiently at the scale of models with 100+ billion parameters. We are the first to quantize such models accurately to low bit-widths. Concretely, our verifiable claim is that GPTQ compresses 175B-parameter models accurately, in approximately the same time that is needed by other PTQ methods to compress models 100x smaller: e.g., 3h for 1.3B by ZeroQuant vs. 4h for 175B by GPTQ. For slower PTQ methods, the difference will be even larger.**
> > > >
> > > > Existing non-trivial PTQ algorithms, in their current form, would not scale to models with tens or hundreds of billions of parameters. We provided the following evidence to support this claim:
> > > > * Our extrapolation of runtime in Figure 3 and the corresponding “Runtime” paragraph, showing that such methods would take weeks to months to run.  Extrapolation is necessary since we are unable to actually run these methods at such a scale.
> > > > * All prior work at the scale of models we focus on, in particular ZeroQuant and LLM.int8(), only apply RTN for models larger than 1.3B parameters. Specifically, the authors of ZeroQuant limit experiments for their LKD variant at 1.3B parameters, after which point this method becomes very expensive to run. Please note that ZeroQuant-LKD is less computationally expensive than OBQ or BRECQ.
> > > > Thus, we are unable to provide a direct comparison with non-trivial PTQ methods at large scale.
> > > > We believe that scaling non-trivial PTQ approaches to massive models is an interesting challenge for the community, and hope that our work will motivate further research into this.
> > > >
> > > > 2. **Accuracy comparison with SOTA PTQ methods on ResNets/BERT.**
> > > >
> > > > We believe they reviewer is reacting to the following statement from our rebuttal:
> > > >
> > > > > GPTQ is more than 50-100x faster than any other existing post-training method with similar accuracy
> > > >
> > > > which is seen as an overstatement, specifically w.r.t. the BRECQ comparison at low bitwidth.
> > > >
> > > > This was not meant as such: in fact, the next paragraphs in our rebuttal stated that we believe that the trend of well-tuned BRECQ performing better than GPTQ will likely continue on larger models, albeit at much higher computational cost for BRECQ. Further, in our defense, we note that we consistently outperform the fastest among advanced PTQ methods (AdaQuant), across all settings/models we tried, and that in some cases the accuracies are similar also vs. the other more complex methods.
> > > >
> > > > To address the reviewer’s concern, we have completely revised this claim. Specifically, we have revised our paper carefully, in particular the related work section, to make sure that this claim is clarified, and that it does not appear that we claim SOTA results for one-shot quantization on ResNets/BERT models, which we are not.
> > > >
> > > >
> > > > 3. **Novelty.**
> > > >
> > > > In brief, our claims to novelty come from:
> > > > * The scale at which our algorithm works accurately. In particular, we more than double the degree of compression on the very largest models relative to prior work [LLM.int8() & ZeroQuant; NeurIPS22].
> > > > * The technical insights which enable scaling:
> > > > 1) random order quantization, enabling the design of a new algorithm with $\text{min}(d_\text{row}, d_\text{col})$ better computational complexity than OBQ;
> > > > 2) addressing the practical memory-bandwidth bottleneck of this algorithm via a (non-approximate) blocking & batching technique;
> > > > 3) a different but equivalent Cholesky-based formulation of our algorithm which achieves numerical stability even on extremely large models.
> > > > To our knowledge, each one of these techniques is new, and necessary for good performance and accuracy at massive model scale.
> > > >
> > > > * The quantized inference framework: we are the first work to show end-to-end speedup and accessibility improvements (specifically, running a 175B parameter on a single GPU) on a popular generative setup.

---

> > > > ### Author Response · Authors · 2022-11-17
> > > > **Response Part 2 (Perplexity Calculation)**
> > > >
> > > >
> > > > We believe the large discrepancy in perplexity scores comes from the fact that the evaluation script used by the reviewer was designed for a different model and dataset.
> > > >
> > > > Concretely, we investigated this in detail and found the following two major problems when running `run_clm.py` as-is for OPT + PTB:
> > > >
> > > > 1. Due to the way encoding and sample merging is implemented, some extra <start>-tokens will be inserted in between text segments, which seems to confuse OPT models (BLOOM has no such tokens hence this issue does not occur).
> > > >
> > > > 2. When concatenating sentences of the raw PTB dataset, no sentence separators will be inserted, which has a huge impact on model performance (this affects both BLOOM and OPT).
> > > >
> > > > Both of these problems can be fixed by replacing a single line of code. Specifically, line 409 should read:
> > > >
> > > > ```output = tokenizer('\n\n'.join(examples[text_column_name]), return_tensors='pt')```
> > > >
> > > > This will resolve problem 1 by jointly encoding the concatenated text and problem 2 by adding “\n\n” separators.
> > > >
> > > > Notice that these are recommended to be added in between samples by at the address you linked https://huggingface.co/docs/transformers/perplexity.
> > > >
> > > > We emphasize that this fix merely properly adjusts the preprocessing of some particular sample script written for a different model & dataset; the core HuggingFace evaluation code, which is indeed used by many researchers, remains unchanged. After this small change, the following command produces results that are almost identical to our reported numbers. Specifically:
> > > >
> > > > ```python run_clm.py --model_name_or_path facebook/opt-125m --do_eval --dataset_name ptb_text_only --block_size 2048 --output_dir tmp --per_device_eval_batch_size 1 --fp16_full_eval```
> > > >
> > > > This yields 27.67 vs 27.55 for OPT-125M/PTB with our evaluation (for OPT-2.7B, it gives 15.19). Without the fix, the script reports 124.49 PPL for this model/dataset combination, which is extremely high.
> > > >
> > > > As orthogonal validation, please note that our baseline perplexities are in line with those from previous work, e.g. LLM.INT8().

---

> > > > > ### Author Response · Authors · 2022-12-02
> > > > > **Perplexity Validation Follow-up**
> > > > >
> > > > > Dear Reviewer X8QT,
> > > > >
> > > > > We wanted to gently follow-up specifically on this point, and ask if you were able to validate the fact that our perplexity calculation is correct once the script used is adapted to match the model/dataset combination.
> > > > >
> > > > > Thank you!

---

> > > > ### Author Response · Authors · 2022-11-17
> > > > **Response Part 3 (Cross-Layer Dependencies and Usefulness of the Hessian)**
> > > >
> > > > **Full precision model input & cross-layer dependencies (items 3 & 4)**
> > > >
> > > > We agree that integrating the dense model output is a promising idea, both in the context of GPTQ as well as in the context of scaling up other methods; the main point of our initial response was to explain that, based on our experience, this is a non-trivial endeavor and hence a good topic for future work.
> > > >
> > > > Specifically, we fully agree that AdaRound/BRECQ do not suffer from any overfitting *at the scale of ResNets & BERT*. However, we note that both BRECQ (in Section 4.1) and ZeroQuant (in Section 5.1) point out that, for datasets that are small w.r.t model size, jointly optimizing more than a couple million parameters at a time leads to worse results due to overfitting. As a single transformer block of BLOOM-176B is about 100x larger than the entire ResNet50, we think that overfitting for block-wise optimization at this model scale could be a problem.
> > > >
> > > > More precisely, our “potential overfitting” claim was justified by the following experiment:
> > > > We ran GPTQ in a setup where each layer is reoptimized based on the current input in the partially quantized model, and the current output in the unquantized one, before applying GPTQ for quantization. This was significantly more computationally-expensive and more complicated and our validation results tended to plummet, especially at larger model sizes. This suggested overfitting, and motivated our current setup.
> > > >
> > > > **Hessian using only input features & calibration data (item 5):**
> > > >
> > > > In our method, the Hessian of the layer-wise squared error is used as a tool for efficiently finding accurate solutions to the layer-wise squared error problem. This Hessian captures the correlations of features within a single layer, which is exactly what makes it possible to compensate for the error of quantizing a weight by adjusting other weights. In turn, this results in a rounding direction different from the initial RTN.
> > > >
> > > > If we ignore the correlations given by this Hessian, and the squared-error Hessian is thus diagonal, then notice that our method will default to Round-to-Nearest choices, since no weight updates are made. Given the massive gap in performance between GPTQ and RTN, this would suggest that the use of the Hessian matrix is beneficial.
> > > >
> > > > More generally, it is well-established [Nagel et al., 2020; Wang et al., 2020; Hubara et al., 2021; Frantar & Alistarh, 2022] that low layer-wise squared error is well correlated with good overall model accuracy. Further, it can also be theoretically motivated from a full second-order approximation of the model as shown by [Nagel et al., 2020].

---

### Official Review · Reviewer_AnMH · 2022-10-24

**Confidence:** 4
**Correctness:** 3
**Technical Novelty And Significance:** 2
**Empirical Novelty And Significance:** 2
**Recommendation:** 6

**Clarity, Quality, Novelty And Reproducibility:**

Clarity
The problem setup and motivation are clear. The paper also provides details about its methodology.

Quality
The writing quality of this paper overall is pretty decent and is easy-to-follow.

Novelty
The paper does not seem to describe prior work's techniques and contributions accurately. The comparison also is not based on a faithful setup of the methodology used by prior work. The main methods such as layer-wise quantization and OBS are also not very novel.

Reproducibility
The paper describes the technique well and should be quite reproducible.

**Strength And Weaknesses:**

Strengths:
- The paper studies a timely problem, which is to reduce the memory consumption of large-scale GPT models without incurring high compression overhead.
- The proposed GPTQ method demonstrates promising results on the two largest open GPT models.

Weaknesses:
- Apart from applying the optimizations to the largest open GPT models, the technical novelty of the proposed method is quite limited. Layer-wise quantization is widely used in prior works, as the authors also mentioned in Section 3.1. The formulation of optimal brain quantization also largely follows [1] and more recent work such as [2].  In fact, Sections 3.1 and 3.2 should be in the background section. The method proposed in Section 3.3 seems to be more like minor tweaks.
- While existing work used INT8 quantization to quantize both weights and activation, such as ZeroQuant[3], they led to actual latency improvement. In contrast, the proposed method uses 3 or 4-bit for weight quantization and FP16 activations, which to the reviewer's best knowledge, cannot lead to real latency improvements (or even lower latency on a single device since the computation still happens at FP16). The end-to-end latency from this work comes more from using a reduced number of devices, which limits its application scenarios.
- The paper claims it as "the first method to leverage approximate Hessian information". This is not true. Hessian information has been used for mixed-precision quantization in prior works such as Q-BERT and Hawq.
- The evaluation is also inadequate. For evaluating the GPT models, the zero-shot, few-shot learning, and fine-tuning results are important. However, the paper primarily evaluates its approach to fine-tuning a few small downstream tasks such as PennTreebank and wiki2. Therefore, the compression might come at a cost of significantly degraded zero-shot and few-shot learning capabilities of these GPT models.
- The training cost saving compared to other methods is not very convincing. Most training time comparison is based on extrapolation rather than doing any actual measurements. The methodology used for extrapolation is also a bit strange because it is under the assumption that the SGD step is constant regardless of the scale of the model. However, there is no evidence that SGD steps need to remain constant and are a hard constraint for prior works. To be more convincing, it would be better to compare ZeroQuant-LKD and GPTQ under the same training budget, e.g., by using the same amount of tokens as inputs or SGD steps in both cases.
- The paper also seems to describe existing works inaccurately. For example, native INT8 quantization (also used in the evaluation) cannot represent SOTA works such as ZeroQuant and LLM.int8(). The former uses token-wise quantitation for activations, and layer-wise kd and the latter also uses fine-grained quantitation for weights to retain model accuracy, which is different from the setup used in this paper's evaluation. LLM.int8() also reports quantization results on OPT175B/BLOOM176B. To be more convincing, it would be better to make a direct comparison with those methods using the exact methods described in those works.

[1] Hassibi et. al. "Optimal Brain Surgeon and general network pruning", 1993

[2] Frantar et. al. "Optimal Brain Compression", 2022

[3] Yao et. al. "ZeroQuant: Efficient and Affordable Post-Training Quantization for Large-Scale Transformers", 2022

**Summary Of The Paper:**

The paper studies the problem of reducing the memory consumption of pre-trained GPT-like models via post-training quantization methods. These models often have billions or even hundreds of billions of parameters, making the compression overhead also a big cost. To address this issue, the paper introduces layer-by-layer quantization methods that find low-bit representations of weight values based on Hessian information. Evaluation of recently released OPT 175B and BLOOM 176B models show that the proposed method can enable low-bit representation of those large models while retaining similar perplexity numbers on several downstream tasks.

**Summary Of The Review:**

The paper studies a timely problem and introduces a technique that reduces the model size of large-scale GPT models. However, the paper largely falls short of accurately describing and comparing with recent advancements in post-training quantization of GPT models. The evaluation is also inadequate in that it primarily demonstrates the results using a few small fine-tuning tasks.

=========================

Post-rebuttal comments:

The authors addressed my concerns in the rebuttal quite well. I increased my score to the positive side to reflect my stance.

---

> ### Author Response · Authors · 2022-11-08
> **Response to Main Concerns**
>
> Thank you for your detailed comments!
>
> We begin our response by addressing your main concerns, as per your review summary, on the comparison with state-of-the-art works and our evaluation in general. Afterwards, we respond to all your detailed comments as well.
>
> **Description of existing works & comparison baseline (item 6):**
>
> Our discussion of related works is focused on the “actual quantization of weights, i.e., their rounding” as this is precisely the aspect of the quantization process we study in this work. We hope the reviewer will agree that, for  models larger than 1.3B parameters, all prior works do this via simple rounding of each weight to the nearest quantized value. We emphasize that the quantization granularity and the choice of the quantization grid are orthogonal: our method can be applied to any grid or quantization granularity. We will extend our discussion to clarify that state-of-the-art works make non-trivial choices in these aspects.
>
> However, we emphasize that this aspect is already taken into account for our baseline:
>
> In all of our experiments we apply asymmetric per-row (= zero-point vector-wise with FP16 activations) weight quantization on the min-max grid. This is exactly the same weight quantization setup used by LLM.int8() and ZeroQuant (we note that ZeroQuant uses LKD only up to 1.3B parameters and that the FP16 outlier decomposition of LLM.int8() concerns only the activations). This means that our RTN baseline actually corresponds precisely to the weight quantization part of those SOTA works. We emphasize that we always apply GPTQ on exactly the same quantization grid/granularity as our baseline. Any improvements come from the fact that our method figures out a better rounding of the model weights to quantized values.
>
> We therefore fully stand by our claims. Nevertheless, we agree that this aspect can be clarified further, and will do so in the next revision.
>
> **Evaluation (item 4):**
>
> All of our evaluations are completely "zero-shot:" we make sure to only use generic text samples from C4 rather than task-specific data for the quantization process. We stress that we do not perform any partial or full finetuning. (Since our method runs on a single GPU, this would not be feasible at this scale.) This setup, as well as the particular tasks we consider, are in line with all recent works on very-large language model quantization [Yao et al., 2022; Dettmers et al., 2022; Park et al., 2022].

---

> > ### Author Response · Authors · 2022-11-08
> > **Response to Remaining Comments**
> >
> > We now respond to your remaining comments:
> >
> > **Technical novelty (item 1):**
> >
> > We suspect there may be a misunderstanding regarding the stringent requirements for applying accurate layer-wise quantization methods to the largest open GPT models, specifically with just a few hours of computation. At the scale of the largest models, our methods provide a speed up over existing techniques of two to three orders of magnitude, i.e., a factor of 100x - 1000x.
> >
> > As you also state, this significant speedup is achieved in a very well-studied area, and thus requires significant new ideas. Our main technical contribution is showing that this can actually be done while retaining competitive quantization accuracy, through a sequence of novel algorithmic insights and techniques. Specifically, we base our results on a new and significantly more efficient algorithm, as well as on new practical insights, regarding both numerical stability and efficient implementation.
> >
> > **Latency improvements (item 2):**
> >
> > Our work focuses exclusively on weight quantization and is thus complementary to the activation quantization techniques in ZeroQuant and LLM.int8().
> >
> > Further, we note that ZeroQuant only provides acceleration for models up to a few billion parameters: as shown by LLM.int8(), emerging outliers break 8-bit activation quantization for the largest models. While LLM.int8() makes almost full 8-bit activations feasible for those models, it currently does not deliver any significant speedup, as is made very clear by the authors themselves.
> >
> > Finally, as we demonstrate, weight-only quantization can actually achieve inference speedups for generative tasks with batchsize 1 via custom matrix-vector product kernels that utilize the fact that the underlying single-token matrix-vector products are memory- rather than compute-bound. We note that almost all the speedup in our experiments is due to this. Specifically, the increased communication cost from using a higher number of GPUs makes up only < 5% of the overall runtime (see also our response to R1/TffF for a detailed discussion of our kernels and the benchmarking).
> >
> > **First method to leverage approximate Hessian information (item 3):**
> >
> > The complete statement in the paper is that GPTQ is "the first method to leverage approximate Hessian information at this scale", by which we clearly refer to extremely large 175 billion parameter models. Q-BERT and HAWQ study models 1000x smaller. We will further clarify this in the next revision.
> >
> > **Training cost savings &  baseline runtime estimation (item 5):**
> >
> > The main point of our method is that with a similar “training budget”, we can accurately quantize models 100x larger than prior methods. Concretely, on a single A100 GPU, ZeroQuant takes 3h for a 1.3B model whereas GPTQ takes 4h for a 175B one.
> >
> > We want to recall that the ZeroQuant paper applies LKD only to models up to 1.3B parameters, to larger models they just apply vector-wise round-to-nearest quantization, which corresponds to our main baseline (as discussed above). Based on the fact that the ZeroQuant authors mention pretraining their models themselves using 128 A100 GPUs (see their Appendix A.2), yet did not scale LKD beyond 1.3B parameters, we suspect that the runtime scaling may actually be significantly worse than our estimates and potentially also bring further unexpected challenges.
> >
> > We want to strongly emphasize that we have kept our runtime estimates for other methods as optimistic and as fair as possible, in particular by assuming constant scaling of SGD steps. We think that the evidence is actually pointing more towards larger models requiring more data and more steps for accurate quantization: more precisely, ZeroQuant uses 100 steps for BERT-base (100M params), 400 for BERT-large (300M params) and 1600 for GPT2 (1.3B params), which suggests linear scaling with parameter count. Our scaling ignores this fact, and is therefore probably undercounting.
> >
> > ***
> >
> > If any questions remain, we are happy to engage in further discussion!

---

### Official Review · Reviewer_TffF · 2022-10-24

**Confidence:** 4
**Correctness:** 3
**Technical Novelty And Significance:** 3
**Empirical Novelty And Significance:** 3
**Recommendation:** 6

**Clarity, Quality, Novelty And Reproducibility:**

The clarity and quality of the paper are good.

The novelty is mostly on accelerating an existing quantization method (OBQ) to make it scalable to a large model setting. I think the technical contribution is non-trivial.

The reproducibility is good since code is provided.

**Strength And Weaknesses:**

Strength:
1. Firstly, the paper is generally well-written and easy to follow.
2. The proposed methods combines empirical insight of ordering in OBQ method, and system optimization like batch processing and numerical precision handling, leading to a holistic solution.
3. Results show that the proposed method can preserve the performance of LLMs at a low-bit quantization (3/4).
Weakness:
1. The author mentioned the method can speed up the inference since "weight matrices can fit into the faster accelerator-local memory". However, for the fp16 model and model parallelism, the weights are also stored in accelerator-local memory (part of the weights per GPU). The throughput can be improved by pipeline parallelism or more advanced parallelism (e.g., [a]). What is the baseline when measuring speed up? Does the baseline include advanced model parallelism techniques? Does the baseline use a competitive implementation like FlashAttention [b]?
2. The authors proposed a quantized-matrix full-precision-vector product kernel. Could the authors discuss how large is the dequantization overhead?

[a] Zheng et al., Alpa: Automating Inter- and Intra-Operator Parallelism for Distributed Deep Learning
[b] Dao et al., FlashAttention: Fast and Memory-Efficient Exact Attention with IO-Awareness

**Summary Of The Paper:**

In this paper, the authors proposed GPTQ, a practical method to perform low-bit weight quantization of large Generative Pre-trained (GPT) models in a post-training manner. GPTQ adapted from existing Hessian-based quantization method and performed modifications to improve efficiency by using a fixed per-row ordering, a faster runtime optimization, and addressing numerical stability issues. Results show that the proposed method can achieve low-bit quantization (3/4 bits) of largest OPT and BLOOM models at negligible accuracy loss.

**Summary Of The Review:**

The proposed method is valuable to the community. It reduces the requirement for people to run a large language model. My only concern is how the baseline is chosen for latency measurement.

---

> ### Author Response · Authors · 2022-11-08
> **Author Response**
>
> Thank you for the review!
>
> We will now discuss all your questions about the benchmarking setup and our kernels in detail.
>
> ***
>
> **Benchmarking setup (item 4):**
>
> First, we clarify our benchmarking setup: we start from the standard HuggingFace/accelerate multi-GPU execution setting, which is also used by the recent work LLM.int8() [Dettmers et al., 2022]. This splits the model by putting a subset of consecutive layers on each GPU. We emphasize that, in this setting, the communication costs are very minor, i.e., < 5% of the overall runtime, even when splitting the model across 8 GPUs.
>
> Our speedups come from the fact that quantized matrices occupy less memory, meaning that memory-bound computations, such as the matrix-vector products that occur for generative inference (with one token at a time), which is the use-case we focus on, have to access less GPU memory overall and are thus considerably faster. More specifically, with a single slow 32-bit main GPU memory read, our 3-bit kernels can load more than 10 weights into a register. While these compressed weights still need to be unpacked for actual computation, this happens with just a few operations performed fully within registers and thus incurs only minor extra compute cost but no additional, much more expensive, main GPU memory accesses. In contrast, standard FP16 computation requires less compute (as no unpacking is necessary) but can only load 2 weights per 32-bit memory read, meaning that it has to perform > 5x more memory accesses, which are expensive for powerful modern GPUs.
>
> Our setups for FP16 and 3-bit models only differ in whether the matrix-vector products are performed in standard fashion or using our compressed kernels described above. This means all overheads incurred by the HuggingFace harness, attention and other non-quantized operations such as residual connections or layernorms are exactly the same between implementations. Consequently, our quantized models should benefit from advanced parallelism or faster attention algorithms just as much as our baselines. We just stuck to standard model splitting for simplicity, following prior work.
>
> Lastly, the sentence about "weight matrices can fit into the faster accelerator-local memory" describes a different setup where not enough GPUs are available to fit the FP16 model and thus weights have to be sent from CPU to GPU memory during inference, which is extremely slow. We agree that this paragraph may be a bit confusing to the reader, since it mixes discussion about different scenarios. We will rewrite it for more clarity.
>
> **Dequantization overhead (item 5 in your review):**
>
> As explained above, we propose kernels specifically for generative inference (with one token at a time) where most operations are matrix-vector rather than matrix-matrix products. For these memory bound operations, our kernels don’t incur any overheads. They are in fact substantially faster than standard FP16 computation: e.g., 2.8x faster on an A100 or 5.2x faster on an A6000 for the FC2 layer of OPT-175B.
>
> For non-generative and large-batch applications, operations may be compute- rather than memory-bound and the technique above is not directly applicable. Instead, one could simply decompress the matrix before performing the corresponding matrix-matrix calculations: this takes ~1.5ms on an A100 and ~3ms on an A6000 compared to 76ms/365ms for the subsequent OPT-175B FC2 layer computation with batchsize 16 x 1024 tokens. Hence, for such applications our methods significantly reduce the required number of GPUs at very little computational overhead. This is similar to recent work [Dettmers et al., 2022], but our work achieves a 2.5x higher compression rate (3bit vs 8bit).
>
> ***
>
> If any questions remain, we are happy to engage in further discussion!

---

> > ### Comment · Reviewer_TffF · 2022-12-06
> > **Reviewer Feedback**
> >
> > Thanks for the clarification! I can see the decoding/dequantization overhead is relatively small, even under batch processing or the context stage.
> >
> > The per-token generation stage is high membound, and techniques like op fusion (as in FlashAttention) help to reduce the memory access (at least for activation, I guess, not sure how much for weights). Could you compare the speed with FlashAttention (or other frameworks with op fusion like FasterTransformer) to see how much the op fusion helps? And discuss if the proposed quantization scheme can be incorporated into such frameworks. Thanks!

---

> > > ### Author Response · Authors · 2022-12-08
> > > **FlashAttention Comparison and Framework Integration**
> > >
> > > Thank you very much for your response!
> > >
> > > ***
> > >
> > > **FlashAttention, Layer Fusion & Overheads:**
> > >
> > > Following your suggestion, we performed additional timing experiments with the highly optimized FlashAttention framework, applying a better attention algorithm as well as various layer fusions. Additionally, we also compared the total time spent in matrix-vector products (mat-vec) with the time spent in all other operations (overhead).
> > >
> > > | OPT-175B/A100 | Precision | Total | Mat-Vec | Overhead |
> > > | --- | :---: | :---: | :---: | :---: |
> > > | HuggingFace | FP16 | 230ms | 216ms | 14ms |
> > > | FlashAttention | FP16 | 220ms | 216ms | 4ms |
> > > | HuggingFace | 3-bit | 121ms | 105ms | 16ms |
> > >
> > > While the FlashAttention framework is, as expected, able to reduce the overheads caused by attention, layer-norms, etc. substantially (from 14ms -> 4ms), it only has a minor effect on the overall runtime (< 5% speedup) for generative inference, which is dominated by the matrix-vector products that are fundamentally limited by the GPU’s memory bandwidth.
> > >
> > > Generally, faster attention algorithms are most useful during *training*, which is typically done with large batchsize and sequence length. This scenario is the focus of the FlashAttention paper. Moreover, optimizations like layer-fusion are most critical on *small models*, where overheads are relatively more significant. In contrast, we study *extremely large models*, whose runtime is dominated by the linear layers and perform *generative inference* which requires only linear rather than quadratic attention in each iteration due to the standard key-value caching optimization, see e.g.: https://huggingface.co/docs/transformers/model_doc/opt#transformers.OPTModel.forward.past_key_values.
> > >
> > > **Framework Integration:**
> > >
> > > As our kernels affect only the underlying matrix-vector products, they could be integrated into any inference framework: one would essentially have to replace the parts where standard FP16 matrix-vector products are performed (e.g. in fused layer implementations) by our kernel code.
> > >
> > > To reinforce this point, we note the recent work by Google made public 1.5 months after our submission [https://arxiv.org/abs/2211.05102 - Pope et al., 2022], which validates that the speedup benefits of weight-only quantization (see Section 3.6) indeed also extend to extremely optimized inference frameworks with highly advanced parallelization. This framework would benefit significantly from our GPTQ algorithm, as it would allow them to also work with accurate 3/4-bit models, rather than just 8-bit.
> > >
> > > ***
> > >
> > > Thank you again for your useful comments and questions! We will integrate additional discussion on those topics into the next revision of our paper.

---

### Author Response · Authors · 2022-11-08
**Author Response Overview**

We want to thank all the reviewers for their useful comments and suggestions!

We summarize our responses to the main common reviewer concerns and questions raised in this general reply. Below, in the individual replies, we provide full-in depth explanations and answers to all specific questions.

***

**1. Questions about our baseline and evaluation.**

We emphasize that our quantization baseline is identical to the schemes applied by the state-of-the-art works on very-large-model quantization, namely LLM.int8() and ZeroQuant. Specifically, we implement row-wise zero-point weight quantization. We denoted this baseline generically by “round-to-nearest (RTN)”, as this is the basic procedure applied by these works at the scale of large models. (As we discuss in the individual replies,  ZeroQuant uses LKD only up to 1.3B parameters.)

Further, GPTQ is always applied on exactly the same quantization grid and granularity as this baseline. Accuracy improvements come solely because of better weight quantization assignment.

We also emphasize that our evaluation is always fully zero-shot: no task-specific data is used during quantization. Moreover, it does not involve any finetuning with task-specific data. Our method never takes any model gradients, and runs on a single GPU. This evaluation mode as well as the tasks we considered are fully in line with all existing works on quantizing extremely large models, namely [Yao et al., 2022] (ZeroQuant), [Dettmers et al., 2022] (LLM.int8()) and [Park et al., 2022] (nuQmm). Finally, we detail our perplexity calculation in the individual response, confirming that it corresponds to standard practice: in particular, as can be seen in the code we have provided, we followed the very resource pointed to by the reviewer.

**2. Questions about speedup measurements.**

We confirm that our speedups are obtained relative to a standard HuggingFace/accelerate (https://github.com/huggingface/accelerate) setup also considered by the state-of-the-art prior work LLM.int8(). Here, the model is split by putting subsets of consecutive layers on different GPUs.

Our speedups do not come from reducing communication cost: in this setup, the total communication overheads are minimal, i.e., < 5% of the total runtime. Instead, our speedups result from removing the memory bandwidth bottleneck of individual matrix-vector products with massive weight matrices, which is significant even on powerful A100 GPUs.

**3. Concerns about the novelty of our method.**

From a practical perspective, at the largest model sizes GPTQ delivers speedups of 100x - 1000x relative to state-of-the-art post-training quantization (PTQ) methods, while maintaining competitive accuracy. We find this remarkable, and in our experience speedups of 100x - 1000x cannot simply be achieved by tweaking an existing method.

Specifically, PTQ is a very well studied area [Nagel et al., 2020; Wang et al., 2020; Hubara et al., 2021; Li et al.; Yao et al., Dettmers et al., Frantar & Alistarh, 2022]. Yet, only one paper (ZeroQuant) attempted to devise advanced methods for billion-parameter models, and stopped at an 1.3B-parameter model. Our innovations are motivated precisely because of the massive scale: we propose a new algorithm that is $O(\text{min}(d_\text{row}, d_\text{col}))$ faster (100-1000x in practice for the largest models) than the related OBQ method, a new way of maintaining the Hessian inverse that is numerically-stable for massive layers, as well as an efficient end-to-end implementation.

Further, these efficiency improvements make it possible to, for the very first time, apply highly accurate 3- and 4-bit post-training quantization to extremely large 175 billion parameter models, pushing their compression from 2x, achieved by current methods, to more than 5x, at minimal accuracy loss. In fact, in the revision, we will show that the same technique can achieve 2.5 bits/weight on average with similar accuracy, by grouping, i.e. reducing the granularity at which quantization is applied.

***

We would like to again thank the reviewers for their feedback, and look forward to an interesting discussion period!

---

### Author Response · Authors · 2022-11-11
**Paper Revision Summary & Discussion Reminder**

Dear reviewers,

we would like to thank you again for your useful comments and would like to note that we just posted a paper revision integrating your feedback.

Specifically, it contains the following:

* Detailed clarifications regarding the inference setup (section A.4.2)
* Further details regarding task evaluation (section A.4.1)
* Additional results showing similarly-good accuracy for 2-bit quantization with grouping (section A.3)
* Additional results for zero-shot tasks (section A.6)
* Changes in order to improve clarity on several points raised by the reviews (main paper).

We would be very happy to engage in further discussion about this paper revision as well as our replies in general and would like to remind everyone that the discussion period ends in one week, on November 18th.

With best regards, the authors.

---

### Decision · Program_Chairs · 2023-01-20

**Decision:**

Accept: poster

**Justification For Why Not Higher Score:**

While the paper is a meaningful step forward, it's not a fundamental breakthrough. It also is mainly of interest to a limited audience.

**Justification For Why Not Lower Score:**

Only one reviewer recommended rejection, and I think their concerns were not valid.

**Metareview: Summary, Strengths And Weaknesses:**

This paper proposes a quantization scheme for neural networks, targeting large decoder-only language models. The algorithm is an improvement over OBQ, and involves using a fixed order rather than a greedy order for choosing weights to quantize as well as algorithmic innovations for faster runtime and numerical stability. The resulting algorithm can quantize larger models (100B+ parameters) to low precision (3-4 bits) on a single GPU without significantly degrading the performance, all of which improves over past work. All reviewers but one recommended acceptance. The reviewer who recommended rejection had concerns about the perplexity of the baselines as well as the importance of efficiency when quantizing. The authors explained why the reviewer was misunderstanding the perplexity issues and I side with the authors in terms of the importance of efficiency. So, I recommend acceptance.

My only (strongly) suggested change is aesthetic - the paper calls its algorithm "GPTQ", but the algorithm is not specific to GPT models and in fact was not applied to any GPT models. Note that GPT is a trademark of OpenAI (https://tsdr.uspto.gov/#caseNumber=90092493&caseType=SERIAL_NO&searchType=statusSearch) used to describe their products, not a generic term for a model family. I would highly recommend changing the name of the algorithm - it's currently confusing and misleading.

**Note From Pc:**

if the above contains the word "oral" or "spotlight" please see: "oral" presentation means -> notable-top-5% and "spotlight" means -> notable-top-25%. As stated in our emails, we are disassociating presentation type from AC recommendations